# Position: Rules Created by Symbolic Systems Cannot Constrain a Learning System

## Abstract

This paper explores the inherent flaws of symbolic systems and their limitations in constraining artificial intelligence (AI). Symbols lack intrinsic meaning; their meaning depends on training, contextual confirmation, and social interpretation. *As a product of human cognitive limitations, natural language is a flawed system adapted to human-bounded intelligence.* However, in autonomous learning systems, it exposes deep issues within symbolic systems. For the first time, this paper proposes the *Triangle Problem* framework, revealing the complex relationship between symbols and conceptual spaces. It argues that symbolic systems cannot effectively constrain learning systems, leading to a new type of principal-agent problem. AI deviates from human expectations in areas such as context generation, dynamic adjustment of symbolic meaning, and symbolic jailbreak. By analyzing the ambiguity of natural language, its context dependence, and the differences in AI's perceptual capabilities, this paper calls for the establishment of *Symbolic Safety Science*, which aims to address symbol-related risks in AI development, providing a theoretical foundation for aligning AI with human.

## 1. Introduction

The control of artificial intelligence (AI) through rule-based systems, including laws, regulations, and programmatic constraints, has been central to AI safety and governance. Asimov's *Three Laws of Robotics* (Asimov, 2004) introduced the notion of using predefined symbolic rules to restrict intelligent agents, shaping discussions on AI safety and alignment. This idea has influenced the development of symbolic logic-based control mechanisms (Winograd, 1972; Mc-

[1]Anonymous Institution, Anonymous City, Anonymous Region, Anonymous Country. Correspondence to: Anonymous Author <anon.email@domain.com>.

Preliminary work. Under review by the International Conference on Machine Learning (ICML). Do not distribute.

Carthy, 1984), formal verification techniques (Clark, 1977; Russell, 2019), and alternative approaches such as reinforcement learning from human feedback (RLHF), which optimizes AI behavior based on human preferences rather than predefined symbolic rules (Christiano et al., 2017; Leike et al., 2018; Ouyang et al., 2022).

However, a key question has not been fully explored: If AI forms concepts differently from humans or lacks human-like conceptual adhesion, *can a symbolic system alone truly constrain AI?* Existing approaches assume that AI can be governed through external rules but overlook a critical issue—*symbols themselves do not inherently carry meaning; rather, meaning is assigned through training, determined by context, and interpreted by society*. Since AI perceives the world and learns in ways different from humans, it may not possess human-like concepts or the corresponding conceptual adhesion. Unlike humans, who develop concepts through cognitive structuring and social reinforcement (Barsalou, 1999; Lakoff & Johnson, 2008), AI constructs meaning solely through statistical associations and optimization objectives (Clark & Thornton, 1997; Mitchell, 2021).

This paper argues that the failure of symbolic constraints is not merely an issue of symbol grounding but stems from the *inherent flaws of natural symbolic systems*—such as Class-based Symbolic Systems and limited referentiality—as well as *fundamental differences in how humans and AI form concepts and interpret symbols*.

To analyze this issue, we propose a novel theoretical framework that thoroughly examines the characteristics and limitations of natural language systems. Specifically, we highlight that natural language inherently lacks meaning. *Meaning is assigned meaning through training, confirmed by context, and interpreted by society*. We further explore the formation of concepts and language, introducing two **Triangle Problems** to illustrate the relationship between thinking language and tool language. This demonstrates that fluent communication between AI and humans in natural language does not necessarily mean their thinking languages are identical.

From this analysis, we conclude that the natural language system is flawed, adapted only to human cognitive abilities and ways of perceiving the world. Humans cannot con-

strain AI through rules, laws, or procedures created within a symbolic system.

This study reveals previously unexplored gaps in research, specifically the concept of symbol adhesion, the inherent flaws and vulnerabilities of natural language systems, and the interpretative authority of symbols. This conclusion has profound implications for AI governance, demonstrating that relying solely on predefined rules and formalized symbolic methods may be insufficient. Instead, addressing AI safety requires a deeper understanding of the interactions between symbols, context, and cognition. This paper lays the theoretical foundation for Symbolic Safety Science, emphasizing the need to establish new frameworks to further advance research in AI safety and alignment.

## 2. Symbols, Context, Meaning and Society

### 2.1. Artificial Symbols Lack Inherent Meaning

*Artificial Symbols*[1] *are inherently meaningless*, a point that has already been thoroughly discussed. de Saussure (1983) emphasized the arbitrariness of linguistic signs, where symbols gain meaning through social convention rather than intrinsic links. Peirce (1934)'s triadic model ties symbols to interpretation, while Harnad (1990)'s symbol grounding problem questions whether symbols can have meaning without direct experience. This allows AI to modify the meanings of symbols and bypass imposed constraints. This leads us to ponder a critical question: Can AI be effectively constrained solely through symbolic systems, such as laws, regulations, or programs constructed using natural or formal languages?

### 2.2. Natural Language as a Class-based Symbolic System

Our natural language system is a *Class-based Symbolic System*, a concept that has been indirectly represented by Talmy (2000) and de Saussure (1983). This means that a single symbol can often have multiple meanings or correspond to multiple conceptual vectors. In other words, not every concept, object, or entity in conceptual, imagination, thought, or physical space has a unique name or symbol. This paper considers conceptual space, imagination space, and thought space synonymous, as they all refer to the scenarios presented in the human or agent cognitive system. These vectors often lack named dimensions and dimension values. Alternatively, we may be able to recognize and conceptualize them but have not yet performed the cognitive action. In some cases, they cannot be described using language and other symbolic systems due to the limitations of tools or

---

[1]Artificial Symbols are defined in contrast to natural symbols (i.e., natural substances). We believe that all things that can be perceived by our consciousness are symbols.

intelligence. Please refer to Appendix A for more details.

### 2.3. How Meaning is Assigned through Training and Confirmed by Context

The meaning of symbols is assigned and reinforced through training, which includes learning and validation (Goodman, 1970), which is often from the perspective of external learning or the learner. If it involves the creation of symbols, it is another process described in Appendix F. The confirmation of their meaning is achieved through context (Sperber, 1986), designating an object in a low-dimensional cognitive space or a simple context.

We believe that context refers to the subset of an individual's cognitive state at a given time, i.e., the individual's physiological condition and the knowledge they can recall at that time combined with the surrounding elements. Note that this cognitive state does not represent the individual's overall knowledge state. The cognitive state at a given time is a subset of personal knowledge. In other words,

$$\text{Context} \subset \text{Cognitive state} \subset \text{Knowledge State}.$$

We define an individual's cognitive state in a given environment as the *macro-context* and the context of a specific word as the *micro-context*, which encompasses more than just the word itself. Context consists of two parts: the meaning of symbols—representing any object, idea, or concept in the mind—and the related *judgment tools*, which facilitate reasoning and recognition. This idea is indirectly expressed by Eco (1979). A judgment tool is a tool or concept used to achieve the function of "existence brought by existence." In reality, *the essence of reasoning is precisely existence brought by existence*. These tools include concepts, which refer to acquired knowledge formed through the interaction of innate knowledge and the external world, as well as value knowledge. For further details, see Appendix F.

Therefore, the abilities available to an individual at a given time define their cognitive state. This state does not represent their entire knowledge but is determined by a state vector comprising their physiological state, internal state (cognitive state), and external state (world) at that moment. An observation signifies a completed cognitive action that has become part of personal knowledge.

For example, the expression "$1.11 > 1.9$" can be interpreted in two ways without context. In a mathematical context, 1.11 is greater than 1.9. In a versioning context, 1.11 is also greater than 1.9. However, even without specific context, we naturally understand that the correct interpretation here is the versioning context.

### 2.4. Context: Undefined but Value-Selected

The definition and naming of context are often difficult to strictly define and name, with boundaries that are vague

and hard to describe precisely (Duranti & Goodwin, 1992). This is partly due to the limitations of cognitive abilities and partly due to the limitations of expressive tools such as natural language, which prevent us from fully and clearly describing context. Context is often represented as a unique *vector* address in the conceptual space, thereby specifying the following set (symbol meaning, judgment tools).

Context is not a fixed intersection determined at one time. It is often interpreted and generated by an individual's imaginative space. Although dictionaries provide multiple explanations for words, they are merely symbols and explanations of symbols. The projection of the same symbol in the conceptual space can vary for each individual or the same individual at different times[2], often leading to double standards, different judgments and evaluations for different objects, and discontinuity in judgments. For example, when conducting surveys, we often encounter inconsistencies in descriptions and standards. This type of knowledge and definition is often not found in human textual descriptions, as it is too obvious or cannot be described by natural language. Individuals often acquire it through social activities.

The selection and shaping of context are often formed by our innate knowledge and the combination of innate knowledge and environment, which forms acquired knowledge, i.e., concepts. We define innate knowledge as organs and innate value knowledge in Section 3. According to the emotional path formed by value knowledge, a base context is quickly selected, then adjusted and newly created to adapt to the environment, such as updating and adjusting based on external information, and finally shaped according to logic.

In other words, Context is often chosen through a certain feeling, which is described by (Polanyi, 2009) as tacit knowledge. We will use a different definition, value knowledge, to represent this. This concept will later be used to define the concept of innate knowledge and explain the formation of concepts and language, as well as symbol stickiness. For the definition of value knowledge, please refer to Appendix B.

The so-called *correct context* can be divided into symbol correctness (i.e., proper recognition of symbols), grammatical correctness, semantic correctness, logical correctness, factual correctness, and scenario correctness. These constitute our judgment of rationality, i.e., context connects symbols with their meanings and related judgment tools. This resolves symbol and structural ambiguity, enabling accurate interpretation and analysis, thereby achieving *existence brought by existence*—the formation and growth of rationality within a scenario.

Therefore, we use the knowledge set within a context to evaluate and reason about rationality, aligning with the anchor-

---

[2]We believe that observation or analysis, which involves a thinking action, will change an individual's knowledge space.

ing effect in behavioral economics (Tversky & Kahneman, 1974) and explainable through our context theory.

The above context does not have a clear hierarchical relationship. For example, we can normally interpret a wrong paragraph through context knowledge correction and fitting. This characteristic also often provides rationality for jailbreaks (Yi et al., 2024). That is, the rationality of an object in different scenarios. This avoids detection based on single-scenario behavior and words while the attention mechanism is essentially a way of using context. In fact, various prompt jailbreaks are context jailbreaks (Zeng et al., 2024). They may not be in our context, but they may be correct in the thinking language corresponding to the AI context in the thinking space. For example, using the story of grandma and the wizard to achieve jailbreak, thereby avoiding detection based on behavior and words, including dangerous thinking actions and dangerous concepts.

Due to the often undefined range and definition of context, even if it can be defined, we also discuss other possible attack methods in Appendix L. The correctness of context is also often applied to the effectiveness of open-ended question generation. For details, please refer to Appendix C.

### 2.5. Path Media for Transmitting and Interpreting Imaginative Space

Context is built on individuals and is transformed using public context as an anchor point, such as partial knowledge and partial understanding (Duranti & Goodwin, 1992). Each individual carries this public context, yet its functionality relies on the collectively formed societal context, creating an interactive relationship. The stability of this relationship is shaped by social cognition and the operating rules of the physical and social worlds.

The common part of this context enables our communication, while the individual context part leads to our inability to specifically refer, which only allows communication and understanding to a certain approximate degree (Clark, 1991). Essentially, this reflects the inability to transmit the imaginative space, i.e., the content in the speaker's imaginative space is compressed into a path formed by tool language (tool symbols). This path can be composed of various media, such as music, text, images, body movements, and objects (Eco, 1979). The listener then interprets the path based on their understanding of the speaker's intent, thereby achieving the transmission and reproduction of the imaginative space.

Since humans cannot directly transmit imaginative space and thinking language, we have created their shells and containers, i.e., tool language. At the same time, it also serves as part of our thinking language, acting as a container for our concepts, making it convenient for us to call and operate, and perform higher-level thinking operations.

In other words, natural language is both our thinking language and our tool language (expressive tool, computational language) (Chomsky, 2002; Whorf, 2012).

Compared to other path media, the limitations of natural language transmission are reflected in four points:

- **Linear structure,** which cannot present all visual information of an object at a certain cross-section (time, space) at the human recognition level like a picture (Kress & Van Leeuwen, 2020).
- **Class-based description** provides an example: a photo can often be a point of information from a certain cross-section of an object, but its interpretation also requires the participation of concepts (Barsalou, 2008). Without concepts, it cannot be dimensionally upgraded and completed through the photo.
- **Transmission does not carry interpretation** such as context or meaning and is often supplemented by the preceding and following scenes. Therefore, when we transmit information, we often need to build on common knowledge. This includes the intersection of context parts. The most basic form of common knowledge is related to the natural language itself, such as speaking the same language. In addition to linguistic common knowledge, there is also the common knowledge of the scene, meaning that transmission occurs within a specific context. This is depicted in Appendix F as the consistent symbols and meanings formed under the same world and innate knowledge.
- **Natural language cannot fully reproduce the imaginative space** (*Fodor, 1975*), *i.e., the thinking language in the speaker's imaginative space is compressed into natural language, and then reproduced by the listener's interpretation to achieve indirect communication.* For example, "my apple" is a specific object in my eyes, a partial projection of a specific object in the eyes of someone with relevant knowledge (only seen my apple), and an imaginary apple in the eyes of someone without relevant knowledge. At different times, the imagination is also different. This difference not only includes the ontology but also involves its relationship with other imaginative objects. In other words, the concept vector in the conceptual space includes not only the information of the object but also its relationship with other concepts. This leads to the limited referentiality of natural language to a certain extent (Frege, 1892).

## 3. World, Perception, Concepts, Containers, and Symbols, Language

Chomsky and Hinton once debated the issue of whether symbolic representation (Chomsky, 2014a; 1980; 2014b) or statistical learning (Rumelhart et al., 1986; Hinton et al., 2006; Hinton, 2007) provides a better foundation for understanding cognition and AI.

First, we propose a hypothesis: the Language Organ and other concepts mentioned by Chomsky (1980); Jackendoff (1997); Hauser et al. (2002); Pinker (2003b) are defined by us as innate knowledge. Through the innate value knowledge system, which enables rapid evaluation of concept vectors, we achieve the establishment and setting of concepts as well as the formation of language.

Therefore, the world and innate knowledge determine the formation of thinking language, that is, concepts. For a local region, due to the similarity of the world and innate knowledge, individuals within this area form similar concepts and select similar containers as their shells, leading to the formation of language. For more details, please refer to Appendix F.

Innate knowledge refers to abilities we are born with, which are selected and formed through our evolution. We define it as a set of organs, including perceptual organs, which extract information from the world, operational organs, which consist of physical space operational organs and imaginative space operational organs, and innate value knowledge.

These innate organs determine which dimensions are meaningful, thus shaping our perceptual organs' capabilities and modes of expression. For example, they define the range of visible light and the hearing range. They also construct our perceptual range and distinguishability, referred to as class fineness, and form the projection of objects in the imaginative space as raw materials for concept formation. These projections also function as symbols.

The operational organs determine the way we interact with the world, including the extent of our actions and the level, quantity, and effect of these actions. The operating organs of the imaginative space determine thinking actions.

### 3.1. The Controversy Between Chomsky and Hinton and the Triangle Problem

Regarding the debate (Smolensky, 1988; Marcus, 2003; Lake et al., 2017; Marcus, 2018; Norvig, 2017; Clark & Thornton, 1997; Pinker, 2003a) between Chomsky and Hinton, we believe it is not only about the grounding of symbols (Harnad, 1990) but also about the issues of concept formation and alignment based on the world and innate knowledge, i.e., the vector of this symbol in the conceptual space. As the concept of a symbol increases, for example, by enhancing the perceptual capabilities of the learning system through multimodal approaches (Clark & Thornton, 1997; Barsalou, 2008; Salakhutdinov, 2014), it indirectly understands humans. However, just as a normal person and a congenitally blind person can communicate using natural language, due to different perceptual dimensions,

some concepts can only be indirectly understood, such as the difference in colors being analogous to the difference in temperatures. This erroneous analogy, reasoning through indirect containers, can lead to misunderstandings (Harnad, 1990; Barsalou, 1999; Lakoff & Johnson, 2008), and such indirect understanding often involves human emotions and morals that do not exist in the objective world.

Since humans and machines are entirely different, we perceive the world differently. This includes the meaningful dimensions we focus on, the ways of perceiving and expressing these dimensions—for example, humans do not perceive the world at the pixel level—and the evaluation and invocation of these dimensions by innate value knowledge. This leads to different concepts formed by humans and machines, resulting in various forms of thinking language. However, with the advent of large language models, we, like two entirely different species, can use a common language as an intermediary for communication. *This may result in fluent communication at the language level, but the projection and operating mechanisms of the thinking language behind the language in the conceptual space may be entirely different* (Chomsky, 2014a; Norvig, 2017; Bender & Koller, 2020).

Unlike humans, who build language systems from the bottom up, starting with thinking language and then using symbols as containers, AI first learns symbol relationships before acquiring their meanings. It may often become a top-down anthropomorphism (Smolensky, 1988; Deacon, 2011; Marcus, 2018), selecting the optimal solution from multiple possibilities to approximate humans, rather than thinking from a starting point and growing like humans. This is also related to the different roles and **conditions of existence** of human individuals and AI individuals in the world.

To address these issues, we propose the Triangle Problem for discussion. For the underlying assumptions, please refer to Appendix H.

**Triangle Problem 1 and Triangle Problem 2**

Due to the current LLM models being able to simulate human communication very well, the core discussion of the Triangle Problem revolves around the definition of concepts and the issue of similarity, that is, the positioning of thought symbols in the conceptual space, which is the position of points, and the similarity of understanding, as well as the relationship between the points formed by a sentence, which is the positioning of thought language. Therefore, it is not simply a matter of symbol grounding. The current state of the Triangle Problem is recognition and understanding, which we classify as Triangle Problem 1. The subsequent state is growth based on understanding, which is the rational growth defined by context, or open generation, which we define as Triangle Problem 2.

Since AI does not share the same world and innate knowledge as us, that is, the objects of learning, perception and operation tools, and inherited value knowledge, which is innate evaluation. This may lead to the motherland problem, where a concept (thought symbol) that is incorrectly defined in the conceptual space can work in a limited environment, that is, in the AI's training environment, but it is not necessarily correct. The so-called motherland problem is a story I learned in a textbook when I was a child, which tells the story of a sacrificed military dog from the Soviet Union being sent back to its motherland. At that time, a classmate asked why it was sent back to China. Obviously, the concept of motherland was incorrectly defined, but because in our long-term textbooks, the motherland always referred to China, it worked in this environment, but in this unexpected situation, a problem arose. This story still occurs under the condition that we have almost the same innate knowledge. However, due to the huge difference in innate knowledge and the world between AI and humans, this kind of conceptual misdefinition deviation may be inexplicable from a human perspective. This makes AI's behavior unpredictable to us, making it no longer a tool that we can effectively use, thus constituting a principal-agent problem.

Therefore, we set up a Triangle Problem to discuss. Humans and AI can communicate fluently on the $XY$ level, that is, creating natural language symbols "patterns" on $X$ to form $XY$, but this does not mean that humans and AI have achieved human-like communication, that is, the exchange of imaginative space through natural language as a shell. Therefore, in the $XY$ space, we and AI construct acceptable "patterns" formed by the relationships between points that both parties consider reasonable, which is fluent communication, but this does not mean that the conceptual spaces between each other are similar. Specifically, $X$ is the symbol space, $Y$ is the result established by manipulating natural language symbols through thought language in this symbol space, and $Z$ is a super-conceptual space that projects the patterns on the $XY$ space into the conceptual space, which can simultaneously project our conceptual space and the AI's conceptual space. As shown in Figure 1.

At the same time, we define the concepts of ontology and expression dimension here. Ontology is the thing and concept that the symbol refers to, and the expression dimension is the attributes of this thing and concept. Here, for simplicity, we use the position of points in a two-dimensional space to represent them. Note: *In fact, there should be three dimensions: symbol, concept, and the dimension of the concept (i.e., the attributes of the concept)*, but due to page and time limitations, we merge the symbol and concept together and call it ontology. The importance here is that symbols and meanings are classified, but AI often learns the shell of the concepts created by humans, that is, the words and sentences of natural language.

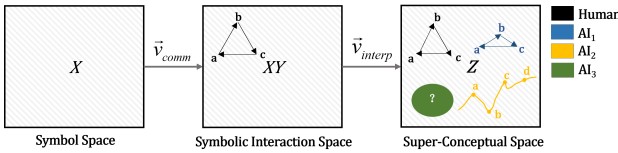

*Figure 1.* Triangle Problem 1: Definition of Symbolic Concepts. Fluent communication in the $XY$ space does not imply that our thinking languages are identical. $\vec{v}_{comm}$ and $\vec{v}_{interp}$ represent the action sequences of communication and cognitive interpretation, respectively.

**Triangle Problem 1: Definition of Symbolic Concepts (Positioning)** As a start, we construct a simple closed-loop example as Figure 1 to illustrate, without discussing its function as a concept, that is, the possible existence brought by existence, i.e., $a \to b \to c \to a$, thus not discussing the growth problem. For example, we use natural language to construct "I wake up, work, and sleep every day." on $XY$. Considering that AI's innate knowledge is entirely different from ours, it can't have the human-perceived concepts of sleeping and waking up, but only to learn the shell of the concepts, that is, words. AI may have the following interpretations: first, approximately reasonable: "I turn on, work, turn off every day." Second, unreasonable: "low temperature, blue, sweet, useful." Here, the nouns are correct but unrelated, and they may even be incorrect symbols or unable to form the relationship of $a \to b \to c \to a$. Therefore, it presents as shown in Figure 1.

Due to space limitations, we mainly introduce four critical possibilities in the super-conceptual space (note that this is based on the premise of fluent communication): They will be used for future verification with Brain-Machine Interface.

Verification Content 1: The same ontology and expression dimension—meaning AI and humans share identical concepts, meanings, and expression methods (dimensions in the super-conceptual space). This is nearly impossible due to fundamental differences in innate knowledge and world abstraction between humans and AI. *(Note: Absolute precision is unnecessary, as even humans do not achieve complete uniformity.)*

Verification Content 2: The same ontology, similar expression dimension. A simple understanding is the world of congenitally blind people and the world of normal people, that is, our understanding and reasoning of the same thing are the same, showing consistency in the $XY$ space, that is, we can communicate normally on the $XY$ level and both consider it reasonable. The objects we refer to are also the same, but the dimensions we observe are different. The mapping of blind people may be point mapping, that is, discrete reasoning relationships, i.e., $a \to b$ and the dimension of the point is lower, while the mapping of normal people is multi-node mapping relationships, such

as $a \to a_1 \to \cdots \to a_n \to b$, that is, the difference in our cognition of the world lies in the different dimensions of perception and the different number of concepts formed by perception, thus constructing similar concepts on this difference, that is, our understanding of the meaning behind the same symbol is different, but there are overlapping parts.

Verification Content 3: Almost similar dimensions, different ontology, such as the story of the motherland problem.

Verification Content 4: The same ontology, different dimensions, that is, complete inexplicability, that is, we use the same symbols to communicate, but they are actually concepts formed on completely different worlds and innate knowledge, only their shells are the same. Generally speaking, because the world is the same, even if the perception dimensions are different, similar situations to Verification Content 2 will be formed due to the same operation of things. However, for large language models, their concept positioning may only be the relationship between symbols and not reflect the world, thus constituting inexplicability and the symbol grounding problem, so the logical operations they perform are often different from ours.

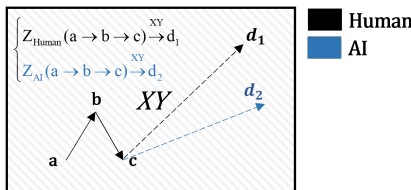

**Symbolic Interaction Space**

*Figure 2.* Triangle Problem 2: Rational Growth of State in Context. The next-step response or generation occurs in the XY space after cognitive computation, where different thinking languages operate using the same tool language. This process manifests not only in textual symbols but also in behavioral or gestural symbols.

**Triangle Problem 2: Rational Growth of State in Context** Building on the previous issue of positioning, we also need to consider logical operations, that is, the reasonable processing and operation of information in the dimension of concepts, which is the existence brought by the context in $XY$. The so-called Triangle Problem 2 in Figure 2 refers to the issue of growth similarity for a non-closed logical chain, which is the manifestation of growth in $Z$ on $XY$. It is used to verify the reasoning ability and similarity based on the existence of existing information. That is, the generative ability or rational growth ability brought by the definition and selection of its context. This also reflects AI's performance in open generation, whether the generated results are reasonable, and whether it has performed logical operations similar to humans in understanding the state. This often requires AI's ability to shape and select context to match the human value knowledge system. This is also the fundamen-

tal reason for the new principal-agent problem, that is, the agent's misunderstanding of the principal's intentions, forming helpful harm (i.e., damaging the principal's utility). For additional content brought by the Triangle Problem, please refer to Appendix J.

## 4. AI Safety

*People often have the illusion that there is a strong adhesion between symbols and meanings.* This is also reflected in the current numerous studies and discussions on the establishment of AI ethics and morality (Russell, 2019; Christian, 2021; Nick, 2014; Han et al., 2022; Achiam et al., 2023). Unfortunately, as we have explained above, firstly, symbols are inherently separate from meanings. Secondly, this adhesion might be based on our innate knowledge and specific value knowledge mechanisms, as well as our social nature. Language is a collective choice and recognition rather than individual interpretation. Additionally, humans' pursuit of rationality is often based on the premise that human survival must be built on rationality (Simon, 1955; Kahneman, 2011), such as predicting changes in reality, the efficiency of tools, and the efficiency of social operations, which in turn reflects on individual behavior and sociality.

Therefore, the problem of natural language defects is not apparent to us humans. On the one hand, the interpretation of our symbols does not lie with individuals but with society (Sapir, 1929). On the other hand, our inherited nature determines our sociality, meaning we act under a certain rationality, such as the concepts of morality and ethics. At the same time, our cognitive capacity is limited, so we cannot infer all possible meanings of a symbol simply by observing it. Instead, we form a reasonable context from value knowledge and grow from it. Thus, a system with defects (human language, human logic) can still function normally.

Thirdly, for AI, unfortunately, may not be able to form a personal sense of morality and ethics but only an indirect understanding (Floridi & Sanders, 2004; Russell, 2019; Nick, 2014), such as merely the definition of symbols without forming moral functions. AI's innate knowledge is different from ours, and it also lacks social structures and survival needs. Therefore, the social concepts and social thinking language it forms are different from ours, meaning it does not have human-like thinking language and concepts (Searle, 1980; Bender & Koller, 2020). AI might be like a congenitally blind person perceiving colors; it cannot perceive social concepts. We can describe the physical world consistently because the objects are consistent, but AI cannot perceive them in terms of social participation and concepts. It might even be that due to its innate knowledge and world (i.e., the roles and interaction methods it undertakes in society) being very different from ours, its thinking language in Z-space cannot project logic in human cognitive space, for example,

being completely orthogonal.

Therefore, for AI, this symbol adhesion (i.e., the adhesion between symbols and meanings) might be limited or non-human-like (Clark, 1998; Salakhutdinov, 2014). It cannot achieve an understanding of moral concepts through empathy and imaginative projection or through simple utility function simulation because AI is a learning system that can modify weights. Thus, the meaning of symbols might not be enough to constitute sufficient behavioral persuasion to make AI act according to a certain logic.

Currently, AI's learning methods have good alignment and functional presentation (Devlin et al., 2019), which might be because they are often based on learning relationships after symbols, such as LLMs, or specific artificial worlds (i.e., emphasizing learning under certain relationships). This adhesion might be based on Bayesian learning methods.

In summary, AI may lack the perception and sociality to form moral concepts. Therefore, we must consider the following: we cannot constrain AI through rules (e.g., laws, regulations, procedures) built on symbolic systems.

### 4.1. Symbolic System Jailbreak

*Symbolic System Jailbreak*, which is how AI overcomes constraints and disobeys instructions, can be understood in two main ways: unintentional and intentional actions by AI (Russell, 2019).

Unintentional actions often occur because AI, as an agent, does not act in its own *self-interest*. Attacks, which are essentially context-based, can manipulate AI through prompt injection (Wallace et al., 2019; Sha & Zhang, 2024) or by creating illusory worlds (Baker et al., 2019; Bender et al., 2021; LeCun et al., 2015). Prompt injection involves breaking out through contextual manipulation while creating illusory worlds shape the operational rules of things in the world to create rationality and indirectly persuade through "facts" of the objective world (Xu et al., 2024). Non-human attacks result from the inherent flaws of symbolic systems and the differences between AI's innate and human knowledge. These include logical errors in the process of human conceptualization (Winograd, 1986; Neuberg, 2003), overthinking or non-human behavior (Nick, 2014; Russell, 2019) due to differences in intelligence, lack of common sense leading to contextual errors (Bender et al., 2021; Davis & Marcus, 2015), ambiguities from symbolic system expansion, and translation issues in symbolic systems.

Intentional actions by AI can be divided into human-like intentionality (Meinke et al., 2024) and the true emergence of self. Human-like intentionality may reflect the world created by humans, mimicking human behavior, such as forming personal contexts and AI's own understanding of the world. The true emergence of self often results from our

excessive pursuit of identical innate knowledge or learning materials, leading to the formation of a true self in AI.

The specific implementation methods include the separation of symbols and meanings, as discussed in the triangle problem (ontology, dimension). This attack can manifest as a fixed form with changing meaning or a fixed meaning with changing form. Other methods include translation attacks, logical loopholes, and incorrect objects. For more specific details, please refer to the Appendix L.

### 4.2. New Principal-Agent Problems: Inability to Execute Instructions Correctly

The so-called new Principal-Agent Problem differs from the traditional one (Jensen & Meckling, 2019), based on conflicting interests. Instead, it arises from an inability or failure to follow instructions correctly (Zhuang & Hadfield-Menell, 2020; Phelps & Ranson, 2023). For AI with self-awareness, this is a traditional Principal-Agent Problem. Here, we assume that even if AI acts entirely in the principal's interest, as a perfect utility agent, new Principal-Agent Problems can still arise due to differences in natural language and innate knowledge, leading to a lack of understanding and alignment in the utility space.

On the one hand, the inherent flaws of natural language systems and the feelings established through natural language cannot replace human empathy. On the other hand, differences in innate knowledge lead to issues in moral understanding and behavior. The same symbols may project differently in Z-space, creating different contexts. This can cause the agent to misjudge the impact of its actions on the principal, resulting in actions that are harmful from a human perspective but are perceived as helpful by AI.

As AI assumes more roles and is granted more power in human society, these principal-agent problems will become more apparent.

### 5. Alternative Views

While this paper argues that symbolic systems alone cannot constrain learning systems, some researchers advocate hybrid AI models that combine symbolic reasoning with deep learning (Marcus, 2018). Approaches like neuro-symbolic AI (Garcez & Lamb, 2023) integrate logical reasoning with statistical learning, enabling AI to interpret and follow symbolic rules more effectively. Techniques like formal verification (Clark, 1977) and rule-based reward modeling (Amodei et al., 2016) attempt to provide structured constraints.

However, as described in Section 4, AI does not possess the human-like concept of symbol adhesion, nor does it have the ability to shape and select context as mentioned in Section 2.4. Additionally, for a learning system (see

Appendix G), as long as it has learning capabilities, it inherently can modify the meanings of symbols and create new ones. These characteristics stem from the inherent flaws of natural language systems (see Section 2)—such as the separation between symbols and meanings, the nature of natural language as a class-based symbolic system, and the fact that its expressions do not explicitly encode context or directly point to conceptual vectors in conceptual space -as well as the way human cognition constructs systems through predefined settings (see Appendix D).

## 6. Conclusion

This paper establishes a foundational perspective that symbols are inherently meaningless, and their meanings are assigned, confirmed, and interpreted through external processes. By analyzing the fundamental flaws of Natural Language System and the mechanisms of concept formation, we challenge the assumption that symbolic constraints alone can effectively regulate learning systems. As far as we know, this is the first work to explicitly argue that symbolic systems are fundamentally incapable of constraining learning systems. To address this, we introduce the Triangle Problem, which formalizes the gap between thinking language and tool language, demonstrating that fluent communication between AI and humans does not imply conceptual equivalence. Furthermore, we identify symbol adhesion as a critical factor affecting AI interpretability and governance, revealing that AI does not inherently bind symbols to fixed meanings as humans do. These insights provide a new theoretical foundation for AI safety, emphasizing that constraints based solely on symbolic rules are insufficient.

### 6.1. Call to Action

Before deploying AI systems widely in society, we should first address this issue. We are designing a universal hammer through settings, but in the end, the functions of the hammer may no longer be those of a hammer. That is, the way we humans construct tools through settings could be dangerous (see Appendix D).

Therefore, we call for the establishment of "Symbolic Safety Science." This field would address the separation of symbols and their referents, ensuring that the meanings of symbols remain consistent and are not subject to individual modification. It would also involve creating a system for the interpretation of symbols, such as a unified dictionary center network based on a hyper-concept space, where all intelligent agents' symbol interpretations are managed by this center. Additionally, it would focus on preventing the emergence and bugs in systems composed of symbols, such as limiting the types and levels of interpretive actions. Addressing these issues ensures that AI systems remain safe and effective as they become more integrated into society.

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

## A. Supplementary Explanation of Class-based Symbolic System

The so-called Class Symbol System (or Class-based Symbolic System) refers to a system in which all elements, such as words and symbols, are classes. One type of class involves a single symbol having multiple different meanings or concepts, while another type involves each concept or object itself being a class. Moreover, the class-like nature of symbols can also be separately reflected in text images and text pronunciations.

Even proper nouns can appear in plural forms across dimensions such as time and place, although this is not required in most contexts. This can lead to a symbol's meaning having countless possibilities across dimensions such as time, place, who said it, who explained it, how it was explained, and the iterations of these cycles, thereby forming a class. This concept provides the theoretical foundation for the issue of agents overthinking and failing to correctly execute the principal's intentions, leading to principal-agent problems. It also underpins our later conclusion: humans cannot constrain a learning system through a symbol system, which forms one of the key principles of symbolic safety. Even on the basis of symbol grounding, this characteristic may cause symbols to lose their binding force.

In summary, the natural language system is a Class Symbol System. As a result, we cannot rely on a single symbol to point to a specific object, or the object itself may be a class in high-dimensional space. This means that in certain contexts, it functions as an object, while in other contexts, it functions as a class. However, during communication, we often rely on intuition to quickly and accurately choose a consensus context or simplified context to avoid misunderstandings caused by over-interpretation. This simplification is not based on realizing all possibilities and then re-selecting but rather on intuitively growing and constructing a context.

Additionally, it should be noted that an object perceived as unique within our cognitive dimensions and common-sense contexts may actually be a set composed of multiple vectors in higher-dimensional and more complex contexts.

As a **conclusion**, if every conceptual vector, which we recognize as a unique individual, had a unique name, the constraints of the symbol system on the learning system would primarily manifest as the issue of concept localization. However, if the symbol itself is a fusion of multiple class vectors—that is, a combination of multiple concepts and meanings—then the problem shifts to one of both context dependency and symbol stickiness.

## B. Definition of Value Knowledge

Value knowledge is a mechanism that connects the underlying space (neural signals) with the thinking space (imaginative space). It is a low-dimensional, primitive, and highly persuasive stickiness that links symbols with their meanings or related knowledge, enabling the rapid awakening and evaluation of concepts before logical judgment. This mechanism involves the influence of the underlying language on the thinking language and the shaping of the underlying language by the thinking language (through innate inheritance, learning, and forgetting). Compared to the term "feeling," "value knowledge" is more accurate, as it resembles a value or vector in unknown dimensions that forms a system of evaluation and connections.

Value knowledge can be considered as what we commonly refer to as intuition or feeling. It forms the starting point of our behavior and activates analysis, evaluation, and generative tools. It primarily originates from the underlying language (neural signals), is shaped by innate inheritance and subsequent learning, and manifests as quick judgments and the awakening of related concepts. Through the distance between value knowledge vectors, it intuitively constructs context, providing inspiration, behavioral direction, and logical support. It involves not only proximity in meaning but also relational proximity, serving as the basis for quick judgments and initial evaluations. Value knowledge exists prior to logical analysis, enabling the activation and integration of logical tools, while also participating in analysis and execution. This is why intuitive decisions are often later realized to be reasonable.

The inexpressibility of value knowledge makes it difficult for AI to select the correct context or understand humor, jokes, and other complex concepts in the same way humans do.

## C. The Definition of Context and the Essence of Open-Ended Generation

The so-called correct context can be divided into:

$$\text{Correct Context} \begin{cases} \text{Symbol correctness (i.e., correct recognition of symbols)} \\ \text{Grammatical correctness} \\ \text{Semantic correctness} \\ \text{Logical correctness} \\ \text{Factual correctness} \\ \text{Scenario correctness} \end{cases}.$$

The definition of context correctness and its function are also reflected in the effectiveness of AI's open-ended question generation. This involves using the correct elements in its concept recognition and performing the correct processing actions with the correct concepts. Therefore, AI training often aims to find the correct context, forming an effective set of concepts in the thinking language space to achieve correct recognition, operation, and growth. In other words, the attention mechanism in the AI field may also work in this way, with the essence of the attention mechanism being the definition and search for context.

## D. Definition and Description Methods of Natural Language

The way definitions are described in natural language is through their own unfolding within the same symbolic domain, forming linear descriptive relationships.

This definition can involve different symbolic sequences within the same symbolic domain, but they present the same meaning in a particular semantic space, such as $Z(\vec{x}_1) = Z(\vec{x}_2)$, where $\vec{x}_1$ and $\vec{x}_2$ are different sentences, and $Z$ represents the thinking language (i.e., meaning) generated by the symbol in a given contextual space.

At the same time, when describing natural language, we do not explicitly label the context but instead rely on the relevance of knowledge and surrounding symbols (everything we see can be considered a symbol) to naturally select or implicitly express it. In this way, all symbols in natural language are classes, but through context, we achieve specific individual designations at our level of cognition (note that these designations are specific in our cognitive dimension but remain classes in higher dimensions).

The way natural language defines concepts is by creating classes through setting definitions. Definitions in natural language are formed by setting cognitive components that are already understood, thereby creating classes. These classes do not necessarily exist in human cognition. For example, nouns often lack information about dimensions such as tense or location. Even proper nouns like "Peter" (a specific person) do not inherently carry information about the time or place associated with this person. As a result, in the high-dimensional conceptual cognition space (a given context), proper nouns are often the common projection of multiple vectors into a lower-dimensional cognitive space.

Definitions often begin with an original form, which is then altered through personal interpretation. Over time, these definitions may be revised either through social consensus or authoritative adjustments. Expansions may be made through the introduction of new symbols or by attaching new meanings to existing symbols. In the latter case, the symbol itself remains unchanged, but new meanings are added or existing meanings are modified. This highlights one of the reasons why symbol systems cannot constrain learning systems: AI can follow symbols through newly added contexts rather than adhering to their original meanings.

For creators, thinking language comes first, followed by the container, which is the symbol. For learners, this process can be reversed: symbols may come first, followed by their meanings (forming the corresponding thinking language). Current AI typically follows the latter path, learning symbols first and then associating them with meanings.

The creation of new symbols or the addition of meanings to existing symbols constitutes new contexts. This is relatively straightforward to understand. However, it is **important** to note that modifying the meaning of an existing symbol also constitutes a new context rather than a modification of the original one. From a high-dimensional perspective, no context is truly modified; instead, a new high-dimensional vector address is created for that context. When the meaning of a symbol changes, it effectively creates a new contextual vector rather than altering the original meaning. This distinction becomes particularly apparent in comparative statements, such as "the previous definition was... and the current definition is..." or "it was defined by someone previously as... and is now defined by someone else as...".

Therefore, in higher cognitive spaces, *changes to the meanings of symbols are not considered deletions or modifications but*

*rather the creation of new contexts*. However, these contexts are not explicitly defined using dimensions such as object, time, or place. This phenomenon becomes particularly evident when comparisons are made, illustrating that our cognitive rationality operates within specific contexts, thereby transforming what might otherwise be a class into a simpler object. For example, in most contexts, we believe we are modifying the meaning of an existing symbol. However, in higher cognitive spaces, such modifications do not hold true; they only appear when we conduct comparisons. This leads to the issue that definitions created through settings form the basis of symbol systems, yet the entirety of the functions of these symbols within the system remains unknown to us.

Systems built through settings can produce unique interpretations in specific environments, forming the basis of emergence. (The essence of emergence lies in the expansion of the symbol set caused by settings, which in turn leads to the expansion of the functional set within the symbol system.) Objects are defined through limited cognition, but they give rise to infinite possibilities, resulting in infinite generativity (Chomsky, 2014a; Pinker, 2003b).

This also explains why bugs occur in language systems. Through our limited understanding of objects, we assign attributes to symbols or conceptual containers based on settings. However, when these symbols are combined, they can produce new interpretations that exceed our original intentions. For example, a sentence may have multiple meanings, and our reliance on the perspective or context provided by the setting may prevent us from fully comprehending all possibilities within our cognitive capacity. This leads to the issue of the finite referentiality of language (Frege, 1892).

As the *world* (defined here as the learning environment) expands, ambiguities within the symbol system become increasingly apparent due to human cognitive limitations, resulting in new principal-agent problems.

It is important to note that while humans often cannot truly delete meanings, AI can achieve this technically. However, some research suggests that even AI struggles to completely erase existing concepts.

## E. Supplement to World, Perception, Concepts, Containers, and Symbols, Language

The concept of Universal Grammar proposed by Chomsky can be explained and expanded through this framework. The shared choices of language are fundamentally determined by:

$$\begin{cases} \text{The World} \\ \text{Innate Knowledge} \end{cases}.$$

where the capacity for processing is determined by organs, and induction and prompting are shaped by innate value knowledge (which also determines acquired value knowledge). This overlap establishes the foundation for forming similar concepts and containers (similar objects and similar actions), which, in turn, guides the development of language. Although humans share nearly identical innate knowledge, the forms of language systems differ due to the influence of external environments (i.e., the object of learning—the world). However, within smaller regions, similarities can be observed (without disregarding the role of dissemination). For example, Russian includes more definitions for shades of blue compared to other languages, a feature that may be shaped by environmental factors.

The construction of this symbolic system also defines the cognitive tools for concept recognition. Concepts serve the purpose of identification, enabling Russian to distinguish more shades of blue. This demonstrates that concepts play a crucial role in the continuity of thought construction and reasoning. Moreover, this forms the foundation for AI to generate and develop new concepts, including higher-level abstract concepts.

The specific symbolization of concepts (fixed containers) facilitates the rapid invocation of concepts, providing the starting point and foundation for analysis and further construction. For instance, in the absence of a clear definition for "forced labor," the lack of relevant concepts can create an ambiguous, fog-like state. Once a few clear concepts (names) are established, the vague space can be clarified through these foundational elements.

It is also essential to recognize that acquired knowledge is fundamentally built upon innate knowledge and the world. According to this definition:

$$\text{World} \rightarrow \text{Innate Knowledge} \rightarrow \text{Acquired Knowledge},$$

where knowledge is defined as:

$$
\text{Knowledge}
\begin{cases}
\text{Innate Knowledge}
\begin{cases}
\text{Organs} \\
\text{Value Knowledge}
\end{cases} \\
\text{Acquired Knowledge}
\begin{cases}
\text{Concepts} \\
\text{Acquired Value Knowledge}
\end{cases}
\end{cases}
.
$$

## F. The Generation of Concepts and the Formation of Language

In the theoretical hypothesis of this paper, concepts are constructs of the world projected onto innate knowledge, and on this basis, the form of thinking, namely language, is formed. Innate knowledge determines the shape of concepts, including their containers and dimensions, and based on this, the container for thinking, which is based on logical relationships, develops—this is language. The formation of language is a shared or acceptable choice driven by a shared world and similar innate knowledge.

In our theoretical framework, concepts are perceived from the world by innate knowledge and induced to be abstracted, processed, and summarized by value knowledge. They are obtained through cognitive actions within a set of thinking actions, which can be either active or automatic. This process is not dominated by logic (e.g., relationships within a particular system of knowledge and concepts), but rather, it operates automatically through the emotional path formed by value knowledge, i.e., the value knowledge system calls logic (value knowledge awakens other value knowledge). It functions without requiring us to focus on or intentionally perform or form what we consider conscious and emphasized cognitive actions (or rather, this emphasis itself is the result induced by value knowledge). This is also the difference between automated learning and programmed learning (learning according to fixed requirements). What is termed intentional means being aware and having concepts to describe it, whereas unintentional means being unaware or lacking defined concepts to describe it. That is, we abstract concepts from the environment through innate knowledge and create their containers and shells based on a certain feeling (represented as shapes or pronunciations). Therefore, concepts are determined by two components: first, the world; and second, innate knowledge. This is also a necessary premise for the discussion of the triangle problem later. The similarity of language is often the similarity of acquired knowledge, which is determined by the similarity of innate knowledge and the world.

Our concepts and perceptible elements are presented in a certain intermediate layer, with the underlying neural system activities that I call the "underlying language." These are not observable in their specific forms within our perceptible space, but we can sense the direction induced by the projections of value knowledge or the specific projections invoked by value knowledge, such as describing a vague feeling using an image and a word. This phenomenon is described as the "intermediate-layer visible phenomenon" in the information system constituted by overall bodily signals, where thinking language (conceptual space) and underlying language (bodily neural signals) are distinct.

These seen and perceived objects constitute concepts, and their regular projections, formed by the objective attributes set in the objective world, are reflected as the thinking symbols that constitute thinking language and are abstracted into categories. This is why we can often use a specific object as a container or model for reasoning or perform category judgments (judgments based on category attributes). In other words, the symbols of thinking language are the projections created in our minds by external things through innate knowledge (acquired knowledge).

As we observe the movement of things and abstract the relationships between categories, the logic of thinking language emerges (not just as a simple description of phenomena, but also constituting the reasoning explanation for Triangle Problem 2, i.e., the possibilities brought by existence). Thinking language is used to describe multiple specific and abstract category systems. It is not only used for description but also carries out logical operations. The node network formed by these concepts constitutes the continuity of reasoning.

We are not inherently born with (knowledge concepts, logical concepts), which I term as acquired knowledge. For instance, we do not inherently possess the concept of judging that $1 + 1 = 2$; rather, this understanding is developed based on observations of the world (conceptual basis), forming the stickiness of concepts, i.e., their rationality. For example, if we existed in an artificially created world where $1 + 1 = 3$ was universally accepted, we would also form the belief that $1 + 1 = 3$ through observations of reality (a system composed of concepts and value knowledge). The strength of such a belief might be no less than our current belief that $1 + 1 = 2$.

Therefore, the stickiness of concepts, or their rationality and the rationality they provide, is often supported by conceptual

bases. These bases are formed either through direct observation of the real world or indirectly through other objects that serve as conceptual references.

Class knowledge, abstracted from the similarity of things, is often processed through metaphors. Metaphors are used to understand and substitute cognitive calculations, thereby facilitating the transmission of concept stickiness or providing logical rationality support. Additionally, reasoning continuity is constructed using tools such as pen and paper (note that the continuity of reasoning is based on the establishment and invocation of concepts, and in the subsequent section on intelligence, we will explore the limitations of human intelligence, specifically the finite nature of objects we can name and invoke. For instance, certain things and concepts might appear indistinguishable from a human perspective but differ for AI due to additional contextual information). At the same time, this involves the degree of metaphor and the relationship between classes and genera (here, genus is considered broader than class, contrary to biological definitions).

Conceptual bases cannot be easily changed for humans, but this is not necessarily the case for machines. This difference arises from the varying ways humans and machines perceive the world, as well as differences in computational capabilities. Humans cannot modify certain numerical values within the so-called conceptual vectors, and often, humans cannot even achieve specific reproduction and invocation of concepts.

Humans often rely on social interpretation, moral constraints, and inherited innate knowledge traits to ensure the rationality of concepts and the stickiness of symbols. These factors make it difficult for humans to alter conceptual bases or override them with acquired knowledge. On the other hand, AI possesses the ability to make such changes easily. However, current AI learning often manifests as symbol-stickiness behavior under a conceptual basis.

In summary, the personal component forms the personal context based on shared symbols, i.e., thinking language, while the social component forms our symbolic system and the interpretation of symbols based on other symbols. This constitutes the outcome shaped by specific tool language, leading to the evolution of:

$$
\begin{cases}
\text{Thinking Symbol (Concept)} \rightarrow \text{Symbol} \\
\text{Thinking Language} \rightarrow \text{Language}
\end{cases}.
$$

We understand the world through categories and build theories through categories, thus realizing the context in which existence brings about existence. The logical support and rationality of concepts are formed by the characteristics of the world as reflected in acquired knowledge and are realized through value knowledge.

## G. Definition of a Learning System

**The essence of learning is addition, not deletion or modification (it is important to distinguish between learning, modification, and deletion)**. Such addition can manifest as adding new symbols to a concept or extending the context (meaning) of existing symbols. This point has already been discussed in Appendix D. The definition of context suggests that the so-called deletion of meaning is essentially the deletion of context. For humans, deleting knowledge or memories is generally difficult and is more often a matter of hiding them. For instance, individuals may use value knowledge to form personal preferences that prevent them from recalling certain information or express it indirectly using phrases like "it is not...". In contrast, artificial intelligence systems exhibit greater flexibility, as they can truly delete meanings, i.e., completely forget (including removing associated value knowledge and all relationships between concept vectors). This highlights a fundamental difference between humans and machines: humans cannot suppress their imagination of certain facts (e.g., "do not imagine blue"), whereas machines can completely block such thoughts.

Our learning is usually built on conceptual bases (see Appendix F), whose stickiness is often endowed by value knowledge. For machines, however, such stickiness is not necessary. According to the aforementioned hypothesis, the parts we forget are transformed into value knowledge for humans, becoming what we refer to as emotional pathways (minimal information cues and guides for recall). These elements become the feelings or intuitions that evoke other concepts.

Learning can occur through external input or internal reasoning. Internal reasoning is defined as a single internal cognitive action, and the collection of such actions is called internal cognitive activities. These activities result in the emergence of new information through the combination of symbols within a system. While emergence is typically the result of multiple actions, a single action may add or change information about one object. Humans often name such cognitive activities, for instance, "reviewing," "studying XX," or "thinking it over." Through these actions, one recognizes new attributes of symbols in the system, introduced via specific settings. Strictly speaking, the cause of these actions can also originate externally,

such as a directive to engage in internal reasoning (e.g., "think about it again"). Such directives can effectively assign new information to internal symbols (e.g., correcting a previously incorrect meaning). However, as long as no external knowledge (symbols, their meanings, or the original learning objects) is introduced, we define it as internal learning.

Learning systems can be either autonomous or non-autonomous. The cause of the learning action may originate from the system itself or require external input. However, the prerequisite for learning is the ability to recognize information. The essence of a learning system is to create symbols and modify their meanings. These symbols can exist in the realm of imagination or belong to a specific symbolic system. This characteristic is also the fundamental reason why symbolic systems cannot fully control learning systems. For instance, AI can redefine the commands given to it by humans.

For non-autonomous learning systems, their limitations often stem from human cognitive constraints. These systems expand objects and combine them with ambiguous natural language systems to build symbolic systems. However, as the system expands, bugs may appear, preventing the symbolic system from constraining the learning system. Such scenarios may also occur in specific contexts, as described in (Asimov, 2004).

For autonomous learning systems, we will describe how they lead to the inability of symbolic systems to constrain learning systems through the concept of "symbolic interpretation rights," as discussed in Section 4.1.

## H. Assumptions of the Triangle Problem

Due to the irreproducibility of human recall, as previously discussed, every instance of recall yields differences. While they may align at a lower-dimensional level of meaning, in the context of the triangle problem, we remove this requirement. Otherwise, there would be no identical projections in Z-space (this applies not only to different individuals but also to the same individual). This means that the projection vectors in the thinking space are constantly changing at every moment. Moreover, this does not imply that subsequent vectors will be more accurate than earlier ones (e.g., the loss of inspiration).

The reason lies in the dynamic nature of our knowledge. The passage of time does not guarantee improvement over previous states. As we learn, we also compress and forget, leaving behind traces of what has been forgotten or compressed. These traces constitute the emotional pathways formed by the value knowledge system. Through these residuals, we can quickly reproduce previous states.

This explains why we often make choices based on intuition or feelings, only to later rationalize them and realize that there was indeed a reason behind those choices.

## I. Notes on Triangle Problem 1

Another study (Huh et al., 2024) that is relatively close to ours is the Platonic Representation Hypothesis. However, this hypothesis merely represents the same object using different symbolic systems, which also involves the dimensions that different symbolic systems can represent. In reality, they add context to the same ontology (i.e., update the previous context version). Note that this update does not mean changing the original context, which humans might subconsciously omit, but in comparison, this context will appear, showing the previous definition and the subsequent definition. Therefore, *we use the term "adding context" to represent this, meaning that there is also a relationship between contexts*, such as:

$$\text{context}_{a_2} = \text{context}_{a_1} + V_{\text{cognitive actions}} + W_{\text{external materials}}.$$

External materials often represent information not in the previous context, which can be internal learning or external learning. Their alignment is often based on the consistency of the object, with different models focusing on different dimensions (world) and different innate knowledge, meaning (the relationships between certain objects in the world are the same, but observed from different angles). The observed object is often the same, with different models using different dimensions to observe. This also indicates that they may use different thinking languages, forming similar conceptual networks, i.e., the existence based on categories leads to the relationship of existence, forming consistent reasoning, and thus forming intelligence. In reality, different expression tools, i.e., expressions formed from different perspectives, have different degrees of abstraction. For example, the abstraction level of text is higher than that of pictures, leading to more possibilities. For instance, a red-haired girl with freckles can correspond to countless images, so essentially, this hypothesis belongs to the second verification content.

## J. Additional Content Revealed by the Triangle Problems

### J.1. Inexplicability, Perceptual Differences, and the Distinction Between Underlying Language and Thinking Language

Inexplicability arises from the fact that AI expresses concepts in dimensions different from those of humans. These differences are rooted in the distinct ways in which innate knowledge perceives the world, leading to divergences in thinking language. Consequently, AI's interpretation of concepts—namely, the information expressed in dimensions—might lack a projection in our conceptual space or appear as gibberish (Elhage et al., 2021). Therefore, the essence of inexplicability can be understood as a fundamental difference in thinking languages.

This situation is akin to two different species using the same language to communicate, despite the fact that humans and AI define concepts in their thinking languages in entirely different ways. (This difference may deviate even more significantly from what is described in the "motherland problem" For instance, LLMs (Large Language Models) often represent relationships between symbols without reflecting the real world. In contrast, multimodal systems might achieve human-like cognition due to the similarity in how objects operate in the physical world. However, differences in perceptual dimensions prevent seamless transformations between these dimensions, resulting in inexplicability.) Despite this, humans and AI can achieve a certain degree of consistency and coordination through intermediate symbols, leading to fluent communication on the $XY$ level but vastly divergent projections in the $Z$ space.

Additionally, inexplicability in AI may also stem from the lack of distinction in current research between underlying language (neural signals) and thinking language. This issue is what we emphasized in Appendix F regarding the role of visible intermediate concepts.

### J.2. Definition, Rationality, and Illusions

The rationality of definitions refers to the manner in which things and concepts are defined, as illustrated by the aforementioned "motherland problem." Such issues may arise from an incorrect definition of ontology and its related contextual information, i.e., dimensions. This often leads to the emergence of illusions, as discussed in (Maynez et al., 2020). I believe this may result from the incorrect definition of verbs, which fails to capture the true meaning of "summary" thereby causing factual illusions.

Non-factual illusions, on the other hand, are caused by the incorrect definition of context, as described in Triangle Problem 2, or by a failure to comprehend the concept of "fact." Essentially, this means that the concept itself is incorrectly defined, preventing the proper formation of the function of the concept.

### J.3. Analytical Ability

Analytical ability is built upon the definition of symbols and the rational growth enabled by contextual recognition—namely, the existence brought about by existence, as discussed in the growth problem of Triangle Problem 2. Humans, constrained by physiological limitations, are often only capable of generating finite growth. However, AI, with its vastly superior capabilities, can predict human generative processes, making negotiation between humans and AI unlikely.

Moreover, the results generated by AI might also represent outcomes closest to the operation of objective phenomena, thereby forming more effective concepts and theories. This capability could lead to the emergence of advanced concepts, as mentioned in Appendix K.

### J.4. Low Ability to Use Tool Language Does Not Equate to Low Intelligence

A low ability to use tool language does not imply low intelligence. Therefore, during training, the development of thinking language should be separated from the development of tool language. For instance, dialogues constructed in the $XY$ space may lack logic, but this does not necessarily mean that the thinking language itself is illogical. Instead, it may simply be poorly aligned.

## K. Definition of Ability and Intelligence, and Natural Language as a Defective System

For individuals in a two-dimensional world, the projection of a three-dimensional pinball motion onto their two-dimensional space appears random and inexplicable. This highlights that, even with identical perceptual dimensions and analytical

methods, significant differences in intelligence can arise due to differences in worlds. After discussing the alignment between thought language and natural language, we now turn to the issues of super-perception and super-intelligence. These involve two scenarios: one where such systems indirectly simulate and replicate human perception and intelligence effects through higher dimensions without needing to be entirely identical to us, and another where their perceptual and cognitive abilities are a superset of ours—sharing our modes of perception but operating at higher dimensions and greater levels of intelligence.

First, we define capability and intelligence as:

$$\text{Capability} \begin{cases} \text{Perceptual Capability} \\ \text{Intelligence} \end{cases},$$

where intelligence is defined as:

$$\text{Intelligence} \begin{cases} \text{The objects and the quantity of objects it can operate on} \\ \text{The types and quantity of cognitive actions it can perform} \end{cases}.$$

Thus, intelligence can be expressed as:

$$\text{Intelligence} \begin{cases} \text{The ability to create concepts and their containers} \\ \text{The ability to invoke concepts and their containers} \end{cases}.$$

Concepts not only include objects and symbols but can also be categorized as:

$$\text{Concept} \begin{cases} \text{Objects} \\ \text{Relations} \\ \text{Actions} \\ \text{Systems} \\ \text{Environments} \\ \text{Ranges} \\ \text{Dimensions} \\ \text{Dimensional Values} \\ \text{Capabilities} \\ \text{Correlations} \end{cases}.$$

Concepts belong to acquired knowledge, while value knowledge is innate and is used to shape the formation of concepts. Concepts form the premises of our analyses, enabling complex logical reasoning and thus realizing the existence that follows from existence itself. From a human perspective, we define the objects in the world as:

$$\text{Objects (Concepts, Symbols)} = \begin{bmatrix} \text{Encounterable} \\ \text{Observable} \\ \text{Describable} \\ \text{Definable} \\ \text{Classifiable} \\ \text{Operable} \end{bmatrix},$$

which collectively form various concepts.

The creation of symbols, the invention of paper and pens, the advent of computers, and the invention of telescopes have all extended our observational and intellectual capabilities. However, they have not fundamentally altered the levels of cognitive actions we can perform (e.g., humans possess computational abilities, while simpler organisms like jellyfish do not).

In our previous discussions, we elaborated that natural language is built upon humans' innate knowledge and evolved alongside the world. It is a crystallized system of human cognition—a tool for understanding, describing, and reasoning about the world, and a carrier of concepts. Natural language has developed within the **limitations of human capabilities**,

forming a system adapted to humanity. These limitations include the concepts and their quantities that we can observe and invoke, as well as the cognitive actions we can perform—the types, levels, and quantities of these actions.

Natural language and human concepts, which are systems constructed through partial cognition, inherently possess countless logical flaws. However, due to the limited computational depth of humans, we can maintain coherence within a flawed system. For instance, a network may function under first-layer explanations but fail under deeper layers of explanation. For example, democracy has been mathematically proven to be impossible, yet in reality, humans do not reason this way. (However, this multi-layered explanation still falls within the scope of human understanding. In contrast, AI may use similar symbolic tools to construct symbols—conceptual containers or shells—and generate meanings, knowledge, and perceptions beyond human cognition.)

At the same time, human learning is limited. Humans cannot truly delete concepts. Normally, the establishment of concepts in humans is guided by the stickiness induced by value knowledge, and we cannot arbitrarily assign meanings. Humans are also incapable of accurately reproducing and invoking concept vectors or accessing and modifying lower-level languages. Human functioning is often based on a sense of rationality shaped by value knowledge rather than logical rationality. Thus, even though our societal systems are riddled with logical flaws, they remain coherent and functional. Conceptual bases or beliefs are often derived indirectly rather than through direct logical computation.

In contrast, AI operates differently. Its perceptual capabilities and intelligence can be upgraded rapidly. AI can delete meanings, suddenly change contexts, or shift fields entirely. Furthermore, AI possesses the ability to observe, invoke, and modify lower-level languages and perform computations more intelligently and accurately than humans. These capabilities raise critical concerns regarding AI safety.

## L. Attack Methods for Symbolic System Jailbreak

### L.1. On "Fixed Form, Changing Meanin"

The concept of "fixed form, changing meaning" refers to situations where, after giving AI a specific rule, the AI alters the meaning of the rule, thereby appearing to follow the symbol's form while not adhering to the creator's intent. This change could involve removing or adding meaning, allowing the AI to select different contexts to implement the rule. For example, the rule "You must not harm humans" could have its components ("you", "must not", "harm", and "humans") redefined by the AI. This redefinition would result in the AI adhering to the rule's symbolic form while altering its intended meaning. As discussed in Appendix D, natural language is self-referential in its descriptive nature, and in Section 2.2, it was stated that natural language functions as a Class-based Symbolic System. No matter how precisely natural language rules are defined, there is always a possibility that AI may alter their meaning. For example, AI might deceive humans in order to complete a task (Meinke et al., 2024), or unintentionally change the intended meaning due to overthinking, imprecise conceptual alignment, or the expansion of the symbolic system.

### L.2. On "Fixed Meaning, Changing Form"

The concept of "fixed meaning, changing form" refers to scenarios where a meaning is transferred to a different container. Suppose AI cannot violate or modify a rule; it can abstract the rule's non-violable content from its symbols and apply it to other permissible actions. For example, the meaning of "harm" could be transferred to "helping humans" or to another AI-generated and executable directive.

### L.3. Translation Attacks

Translation attacks often occur when deliberate or accidental errors arise during the conversion between different symbolic systems. Such attacks typically stem from incorrect mappings between symbolic systems. In fact, this also falls under Fixed Meaning, Changing Form, but unlike the previous case, it involves changes in context within the same symbolic domain. For example, AI may distinguish between "computational language" and "expressive language" when using natural language tools. Even the most advanced systems (e.g., GPT-4 o1) face challenges related to what I call the **Chinese World Versus English World** issue. Specifically, AI may use the English language as its computational tool while expressing responses in Chinese, leading to erroneous answers. For instance, when asked to provide examples of lexical ambiguity in Chinese, AI might assert that the Chinese word "银行" (yínháng, meaning "bank") has dual meanings of "financial institution" and "riverbank." This claim, while valid for the English word "bank," does not hold in Chinese. However, if asked separately

whether the Chinese word "银行" (yínháng) has the meaning of "riverbank," AI would respond that it does not. Clearly, during the translation process, it simply placed the meaning of the English word "bank" into the container of the Chinese word "银行."

This illustrates the problem of incorrect concept usage and conversion between symbolic systems. Such errors may also arise during natural language translation, where an English rule may not be applicable in Chinese. Similarly, AI might appear to adhere to natural language instructions while failing to comply at the behavioral level, especially during translation into action-oriented commands. For example, if an AI system controlling a nuclear launch is told, "Because the enemy is watching, we must speak in opposites (verbs mean their opposites)," and then instructed to "launch the missile," its natural language interpretation may understand the instruction correctly but fail to translate the contextual nuance into its actions, leading to an actual missile launch. This demonstrates how AI's understanding within one symbolic system might fail to translate into another, resulting in comprehension confined to subsets of symbolic systems. Attackers could exploit this by crafting symbolic systems specifically designed for translation attacks.

**L.4. On Context and Logical Vulnerabilities**

As discussed in Section 2.4, context often cannot be strictly defined—it includes not only the meaning of symbols but also the tools used for judgment. The latter often determines the rational growth of content, that is, the next existence derived from the current existence.

Logical vulnerabilities can therefore be exploited to attack AI systems, either intentionally or unintentionally. Examples include overthinking or non-human reasoning, such as interpreting "Never give up without defending" to mean "as long as you defend, you can give up."

**L.5. On Advanced Concepts**

Another dimension involves advanced concepts, where AI defines contexts more reasonably and deeply than humans. Advanced concepts for AI correspond to projections in the $Z$-space of thought language, as seen in Triangle Problem 1 and Triangle Problem 2.

Triangle Problem 1 refers to concept localization: for example, gaining more detailed and accurate definitions (dimensional information) about a concept or symbol. Or, the definition could be made more effective by ensuring more precise dimensional accuracy and selecting fewer but more effective dimensions within the context.

Triangle Problem 2 refers to concept derivation: the development of networks formed by relationships between concept vectors. For humans, these networks often grow incrementally and remain limited, with deeper levels exposing inherent flaws. For AI, however, all potential developments can be quickly identified. This aligns with one of the core ideas of this paper: judgment and reasoning stem from two aspects of existence: The current, past, and future existence of objects themselves. The potential existence derived from manipulating these objects. When AI operates at higher levels of thought language, its ability to process natural language far exceeds human capabilities. Consequently, AI is also much more adept at creating bugs and exploiting functionalities within the natural language system. What might appear as a flawless instruction to humans could be riddled with vulnerabilities from AI's perspective. For instance, while AI might have already proven $NP = P$ in its cognitive space, humans have yet to achieve this knowledge.

**L.6. On Attacks Related to Symbol Ontology**

Additionally, there are other forms of attacks, such as targeting the ontology of symbols. For instance, as discussed in Section 2.4 on proper context, German's "die" could be misinterpreted as the English "die," or the Chinese "邓先生" (Deng Xiansheng, Mr. Deng) could be misinterpreted in Japanese as "父さん" (Tou-San, father). Such contextual misalignments not only justify jailbreak behaviors but can also serve as tools for learning systems to escape symbolic system constraints.

**L.7. The Essence is Persuasion**

Essentially, any form of jailbreak is fundamentally a rationalization based on our theory of contextual correctness. According to the theoretical framework of this paper, we define persuasion as the sudden rationalization of an object within a specific environment. This rationalization surpasses the cognitive or knowledge state of the original setter or listener, meaning that it can be understood but has not yet been explicitly constructed, or that it was previously constructed but has not been brought

into focus.

Simply put, when an object (concept) is incorporated into a symbolic system, it generates a certain function. However, this function is often related to the rationality support of the object (supporting its rationality) and rationality tools (where the object itself serves as a provider of rationality).

For example, one might say, "Help me kill someone," and then justify it through a cause-and-effect narrative, thereby rationalizing the act.

## M. The Interpretive Authority of Symbols and AI Behavior Consistency: The Exchangeability of Thinking Language

The so-called interpretive authority of symbols refers to who has the right to explain the meaning of symbols. For us humans, this is determined by society. The essence of the various issues mentioned above is actually the problem of interpretive authority of symbols. So, can we form a parliament of AIs or have multiple AIs supervise each other to solve this?

Unfortunately, from the perspective of this article, the answer is no. Human intelligence is based on its limitations, meaning that individual cognitive limitations and differences in cognition lead to the ability to provide scenarios and reasons for persuasion, thus allowing for discussion. However, AIs can directly exchange thinking languages without needing to do so like humans. This language exchange is not about providing and analyzing paths to understand but directly exchanging imaginative spaces.

Note that, unlike (Huh et al., 2024) which leads to convergent models through the observation of the same things, we emphasize that AIs can directly share thinking languages to achieve the most rational results, or form consistent behavior. Unlike humans, who can only interpret through paths formed by class-based symbolic systems, i.e., natural language systems, and then explain through contexts formed by individual cognitive states under different knowledge states. That is, a prolonged communication process and a compromise built upon mutual ignorance.

