# OpenReview forum: "Position: Rules Created by Symbolic Systems Cannot Constrain a Learning System"
_ICML.cc/2025/Position_Paper_Track — Submitted to ICML 2025 Position Paper Track_

### Official Review · Reviewer_SB8V · 2025-02-25

**Significance:** 1
**Argument Clarity:** 1
**Rating:** 1
**Confidence:** 4

**Questions:**

I don't have any questions that are likely to change my opinion.

**Discussion Potential:**

2

**Paper Summary:**

The paper argues that the symbolic representations of human and AI systems are not well aligned. It decribes a problem called the "Triangle Problem", which I did not fully understand, despite the elaborate (but not very precise) description, but it seems to be that different entities have different underlying groundings of their symbols, a view that does not strike me as novel. The paper also calls for the establishment of "Symbolic Safety Science", essentially the establishment of "a unified dictionary center network based on a hyper-concept space" (cf. last paragraph), which to me sounds quite similar to the key arguments brought forward for the Semantic Web and ontologies.

**Position:**

No

**Position In Title:**

Yes

**Related Work:**

3

**Strengths And Weaknesses:**

## Main Weaknesses:
1. seems out of scope for a machine learning conference
2. the position does not appear to be new
3. I also believe that some of the claims are wrong.

## More detailed Comments

First, I don't think that his paper fits into the scope of a machine learning. It is a very philosophical AI paper, following up on the symbol grounding discussion, which is going on since Harnad's seminal paper of 1990. There is no discussion on any practical or theoretical ramifications for machine learning. I believe that this would already be sufficient reason for rejecting this paper.

I also believe that the argument is flawed. For example, the authors argue repeatedly that AI symbols are meaningless, and not grounded in experience as our symbols are. And they argue that this not only holds for conventional symbols (which are, in a way, arbitrary, as they are also with humans (different languages have different symbols for the same thing)), but also for, as the authors put it, "the positioning of thought symbols in the conceptual space", i.e., for semantic embeddings. However, the latter is clearly not meaningless. While the labels are exchangable, obviously, the semantics of such embeddings is reflected in the relative positioning in the points, and certainly not arbitrary. Thus, AI symbols that are connected to a semantic embedding space, are certainly not meaningless. I would even argue further that the symbols in LLMs are grounded, albeit in a different way as our symbols. While we ground our symbols in multi-modal perceptions, the symbols in an LLM are grounded in a single-modal perception, i.e., their co-occurrences with other symbols.

## Minor points
- in a mathematical context, 1.11 is *not* greater then 1.9, in a versioning context it is.
- I believe the term "jailbreak" is not sufficiently well established in AI in order to be used without explanation. For example, the authors write "For example, using the story of grandma and the wizard to achieve jailbreak, thereby avoiding detection based on behavior and words...", without giving any explanation of what the story of grandma and the wizard is. I don't think that this is common knowledge.
- the paper is in general hard to follow, there is no clear definition of many terms, also not of those introduced in this paper (e.g., the triangle problem). Lots of references to various appendices make it even harder to understand the paper. I am not willing to read an 11-page appendix in order to understand a 8-page paper.
- There are also some strange claims, such as
  - "AI is a learning system that can modify weights". Apparently, the authors assume that LLMs are the only kind of AI.
  - "humans do not perceive the world at a pixel level". I do think that our eyes do just that. And, of course, we can make sense of pixel-level images (e.g., TV, cell phones, etc.), we only need to abstract away the pixels, which is just what a deep network does (of course the details differ).

**Support:**

1

---

> ### Author Rebuttal · Authors · 2025-03-27
>
> Thank you for your review comments. ***You have stated that you disagree with the position of this paper and have expressed your own position***. Additionally, you mentioned that you are unwilling to read the appendix. To minimize your workload, we will respond using the content already written in the main body of the text, and since this content is part of the main body, we will refer to page numbers and line numbers in our response.
>
> # Response 1: *This paper has been strictly written according to the CFP standard and refers to previous similar accepted papers. Please verify.*
> 1. Position: Beyond Personhood: Agency, Accountability, and the Limits of Anthropomorphic Ethical Analysis
> 2. P: Stop Making Unscientific AGI Performance Claims
> 3. P: AI-Powered Autonomous Weapons Risk Geopolitical Instability and Threaten AI Research
>
> # Response 2: ***Regarding your objections and stated position, please confirm if your (Semantic Embeddings) and insights have already been addressed in (line 14, L, page 1, the first 222 English characters)***.
> > Symbols lack intrinsic meaning; **their meaning depends on training**, contextual confirmation, and social interpretation- Abstract line 3.
>
> Therefore, we will introduce the content from page 2 of the paper.
>
> In 2.1, we state that artificial symbols are tools for communication, containers of thoughts, and have no meaning (thought attributes) without any training.
>
> In 2.3, we emphasize that symbols acquire meaning (concept containers) through training. Sec 2 emphasizes the characteristics of natural language, leading to Sec 3 where we explain that natural language is the shell of human thought, the result of the combination of human ability and the world.
>
> Additionally, your view (I would...LLMs are grounded, albeit in a different way as our symbol) **is clearly stated in the main text (line 292 R)**.
>
> ***Conclusion: Therefore, there is no flaw as you mentioned. These issues have been comprehensively addressed and clearly stated in the first 222 characters of the article. Please verify.***
>
>
> MP1: Appear to be an unfinished response.
>
> MP2: We do not provide detailed method explanations (GR). ***For the readers, we cannot guarantee whether they have read or seriously engaged with the text, but we guarantee that we have provided a comprehensive explanation***. We offer an understandable direction in the methods as described in the text, and we provide a complete mechanism based on our contextual theory (Sec 2.4; 4; 4.1; C; L).  We provide a new perspective and offer a detailed theoretical explanation that the essence of jailbreak is the rationalization of context.
>
> MP4:
> # 1. There is no such assumption that we only refer to LLMs. Please read the paper title, Rules Created by Symbolic Systems Cannot Constrain a *Learning System*.
>
> 2. This claim is not strange. It originates from a very basic and well-known article in the relevant field (Barsalou,1999) on PSS theory.**The following papers provide similar descriptions**:
>     1. ImageNet-trained CNNs are biased towards texture; increasing shape bias improves accuracy and robustness.
>     2. Explaining and Harnessing Adversarial Examples.
>
> MP3:
> 1. These definitions are already provided in the main body, as intended in Sec 2 and Sec 3, such as the definition of the Triangle Problem (page 5, line 258).
> 2. ***To facilitate community discussion, we have provided all perspectives and detailed theories to the best of our ability. Therefore, we cannot condense this innovative content that fills significant research gaps into just 8 pages. Under solid citations and investigations, here are our contributions:***
>    1. First to propose this position, along with a detailed theoretical foundation
>    2. First to strictly define the Class-based Symbolic System
>    3. First to strictly define the nature of context and reasoning
>    4. First to strictly define the modification of symbol meanings, which essentially involves adding new contexts
>    5. First to define the functionality of context and provide a complete classification of Correct Context
>    6. First to define Symbol Stickiness and Concept Stickiness
>    7. Improve the definition of the Language Organ and introduce the concept of an Innate Knowledge
>    8. First to define that language is the shell of human thought, and that human logic and natural language form a defective system, adapted only to human abilities
>    9. First to distinguish between thinking language and tool language and provide the Triangle Problem
>    10. First to strictly define the mechanism of concept formation (types of concepts and the relationship between objects and observers in the world)
>    11. First to propose that generativity and jailbreak are essentially issues of defining context
>    12. First to strictly define and propose the new type of Principal-Agent Problems
>
> **Please verify after reading the content of this paper.If doubtful, please provide a full comparison from both full original texts, not LLM**

---

> > ### Comment · Reviewer_SB8V · 2025-04-08
> >
> > I was just made aware that I posted the comments below as an official comment, and that these are not visible to authors. So, I repeat them here again:
> >
> > Comment:
> >
> > It is really hard to follow some of your comments. I restrict myself to some:
> >
> > No, I am not willing to read appendices, nor am I required to do so. If reading the appendix is necessary for understanding the paper, then please submit it to a journal, where there is no bound on the length of submissions and reviewers can take more time to digest it.
> >
> > MP1: My sentence is a syntactically correct and complete English sentence.
> >
> > MP2: Your response does not address the main criticism (that you do not sufficiently well explain your terms)
> >
> > MP4: I retract the comment that you apparently restrict your arguments to LLMs. My criticism, however, was directed towards the statement "AI is a learning system that can modify weights". From the titles of the papers you cite to counter my claim (the publication venues are not given), they all seem to deal with neural networks. My point was that the definition of AI is certainly much broader than neural networks, there is a lot of work in AI that is not about modifying weights or learning (and there are many learning approaches that do not modify weights).
> >
> > MP3: I am impressed by the 12 contributions that you intend to make. I agree that you "cannot condense this innovative content that fills significant research gaps into just 8 pages". I therefore think that this conference is not the right venue for publishing this work.

---

> > > ### Author Response · Authors · 2025-04-09
> > >
> > > Thank you for placing the comment in the correct channel so that I can further clarify.
> > >
> > > R1：The full context is expressed in the main text as follows:
> > > 1. > ***the expression “1.11 > 1.9” can be interpreted in two ways without context.*** In a mathematical context，1.11 is greater than 1.9. In a versioning context，1.11 is also greater than 1.9. ***However，even without specific context，we naturally understand that the correct interpretation here is the versioning context***.
> > > 2. This appears in the main text on page 2，line 99，rather than in the appendix. I believe the original sentence is also a syntactically correct and complete English sentence.
> > > 3. This sentence is used to illustrate that the issue with current LLMs failing to perform accurate mathematical reasoning is not simply due to a lack of concepts，but rather due to instability in the symbolic system caused by incorrect context definition. The deeper issue lies in the relations between concepts (i.e.，conceptual stickiness)，as well as in how context is defined，selected，or reconstructed.
> > >
> > > R2：I am limited to 5000 characters in my response，so you should specify exactly which terms you believe are not explained. It’s not feasible for us to respond to everything by quoting the original text，as in the example above. However，for the terms you mentioned，we have already responded with specific line references. Therefore，we address this issue in [R4].
> > >
> > > R3：You can look up the influence of these papers yourself
> > >
> > > In addition，this paper distinguishes between autonomous learning systems(ALS) and non-autonomous learning systems(NALS)—that is，who modifies the weights—and presents different results for each case.
> > > For NALS，the inability to constrain them is a traditional problem. On one hand，it stems from the limitations of human intelligence: we cannot exhaustively enumerate all contexts or situations. We can only add constraints in predefined forms，and partial understanding of a phenomenon leads to its existence in the system，but without the ability to fully compute all of its symbolic implications. This leads to emergence，and bugs arise from that.
> > > On the other hand，it’s due to the expressive limitations of our symbolic systems—our inability to fully express our intentions.
> > > This also addresses AXMX’s question: “Is ... a symbolic system?” Yes，they are symbolic systems (line 368，G).
> > >
> > > Therefore:
> > >
> > > 1. The inability to constrain ALS is due to organic differences (symbolic stickiness，conceptual stickiness)
> > >
> > > 2. The inability to constrain NALS is due to the expressive limitations of human symbolic tools，and the limits of human cognition itself
> > >
> > > R4：In principle，this may not be something that should be said in a review response，but I feel it must still be expressed. The reason you are a reviewer today is because others before you took the responsibility to review your work seriously—this is a legacy of responsibility. When you see reviewing as a burden or merely a task，please remember: reviewing is not an unrewarded job. It is precisely because earlier reviewers approached their role with care that the peer review process has continued to this day.
> > >
> > > But your review contains only errors—we could not identify a single point that was accurate. Our main paper is written with sufficient clarity for the reader to understand; ***yet due to the limited space allowed，we were forced to quote line 14 on page 1 and line 99 on page 2 just to clarify basic misunderstandings. I do not believe this aligns with your statement that “If reading the appendix is necessary for understanding the paper，then please submit it to a journal.”***
> > >
> > > You referred to our contributions as things we merely "intend to make"—however，the corresponding contributions are already clearly presented in the following sections:
> > >
> > > 2. Sec 2.2; A
> > > 3. Sec 2.3; 2.4; 2.5; K; L.5
> > > 4. D; G
> > > 5. Sec 2.3; 2.4; C;
> > > 6. Line338，Line819
> > > 7. Sec 3; 3.1; E; F
> > > 8. Sec2; Sec3; E；F；G；K
> > > 9. Sec 2.5；3.1；F；J；
> > > 10. Sec 3；E；F；K
> > > 11. Sec 2.4；L.7
> > > 12. Sec 4.2. We emphasize that even if AI functions as a perfect utility projection (i.e.，it has no utility of its own and merely reflects human utility)，TP1 and TP2 can still arise because the connection between humans and AI is established through a symbolic system. Even if TP1 can be resolved，TP2 will still occur due to differences in the world and the organic differences between AI and humans. Our proposed solution is explained in our response to 8rwE[3-2].
> > >
> > > We may add a glossary of terms and a directory of definitions in the appendix, and we will correct some obvious errors:
> > > 1. Fix the typo "Meanin" on line 1068
> > > 2. Abs: <+position> paper
> > > 3. Use bold formatting for the stated position in the Intro
> > >
> > > You may be trying to poison the atmosphere and discourage the authors，but I won’t abandon my principles. As I noted in my rebuttal [8rwE，3.2]，constraint comes from both outside and within. Even without external pressure, my self-respect and sense of shame keep me from indulgence or bullying 一 I will not follow that path.

---

### Official Review · Reviewer_AXMX · 2025-03-13

**Significance:** 1
**Argument Clarity:** 1
**Rating:** 1
**Confidence:** 4

**Questions:**

Everything I can say is in the main box above.

**Discussion Potential:**

3

**Paper Summary:**

I have to admit that I had a lot of trouble understanding the position in this paper.  Below is my best attempt to state something it might be.

Modern AI systems use language with apparent facility.  That fact might tempt us to think that we can constrain these systems to behave in certain ways by telling them things in natural language (e.g., don't go into the living room or don't break windows).  But the fact that the symbolic outputs of these systems often make sense doesn't mean the behavior of these systems can actually be constrained by telling them things.  There are at least two reasons for this.  A superficial reason is that they might not *actually* have the right interpretation of our words due to ambiguity or lack of training.  A deeper reason is that the semantic interpretation of the linguistic inputs and outputs of such a system *has no real causal connection* to the system's behavior.

But what I wrote above is probably wrong, because the official position refers to rules "created by a symbolic system".  I'm sorry.  I think I just fail to understand.

## update after rebuttal
I appreciate the drive and enthusiasm of the authors.  They clearly have a complex, detailed message to get across.  This message seems to be incompatible with the form of a position paper, which should be self-contained, clear, and of general relevance to a substantial subset of this research community.  My sense is that the authors' paper doesn't have these properties and would probably best be published as a long-form journal paper.

**Position:**

No

**Position In Title:**

No

**Related Work:**

3

**Strengths And Weaknesses:**

At the highest level, the idea that you can't ever really be sure of preventing a system from executing some kinds of actions by telling it not to is important, and possibly one that we should study.

It's important to be very clear, though, by what you mean by a symbolic system. Is a learning algorithm that produces hypotheses in the form of decision trees or programs a symbolic system?

The paper is very highly abstract. I think to get the ideas across more clearly, we need some more concrete examples, very early.  I'm afraid I wasn't able to extract the important message from the triangle examples in the paper.

I wish I could offer more detailed insight or helpful critique.

**Support:**

1

---

> ### Author Rebuttal · Authors · 2025-03-26
>
> # Global Response
> I would like to thank the four reviewers for their reading and valuable suggestions. Please forgive me for not addressing you by full titles due to word limitations. First, I plan to summarize the common issues raised by everyone and respond to them through the AXMX summary. I believe some of the reviews and questions might have stemmed from misunderstandings of the paper's intended message.
>
> > Background:
> **The paper aims to address the increasing role of AI in society, where human production, or interaction with the material world, is increasingly mediated by AI. In this context, we must consider how to constrain AI’s behavior so that it does not become just an experimental problem, *but a conceptual update problem in real-world activities*. This is not a position paper on the technical environment and theory. In this context, we discuss how AI, due to its learning characteristics, will form its own concepts and modify the meaning of symbols during its interactions with the world (a process that may not involve human participation, such as observations from a third-party perspective or summaries of efficiency based on social identity)**. This is due to the differences in perception dimensions and cognitive abilities between AI and humans (Sec 3). This negates the position that focusing only on the rationality of symbolic representations is sufficient (8rwE). Symbolic consistency-XY does not equate to consistency in thought-Z, and as AI accumulates experience in its environment, this deviation will grow, leading to a reinterpretation of the meaning of rules formed by symbols. Under our investigation (SB8V), we are the first to propose this idea and rigorously and comprehensively support it. Additionally, we are the first to introduce the concepts of Symbolic Stickiness (the stickiness between symbols and meanings) and Conceptual Stickiness (the stickiness between concepts) to describe the nature of this evolution in the interaction process, and we have provided validation directions through the Triangle Problems. Therefore, this is not a simple issue of SGP (Sec 3.1).
>
> **Thus, the focus of the issue is on what kind of agency and to what extent it should be deployed (6.1 Call to Action), which is also what we aim to trigger in the community. Conclusions based on experimental environments cannot guarantee the safety of real-world agency processes**.
> 1. What kind of agency level? This level of agency comes from the extent to which we give AI more capabilities to interact with the real world, as well as the social roles and permissions we assign to AI.
> 2. What kind of constraints should be applied? This is not something we should provide; our position is to explain why it is not feasible. On the one hand, we want the community to discuss it; on the other hand, we plan to address this in our next paper. You can refer to the discussion on split-brain syndrome[3.] and communication between AI[7.] in the response to 8rwE, as well as the four verification contents and examples based on brain-machine science presented in the paper (line 287, AXMX).
> 3. How should these constraints be designed? (Our rules are based on symbols.) How can they be converted into relevant constraint measures to prevent the separation and modification of symbols and meanings and avoid loopholes caused by human limitations?
>
> **This paper argues that the key lies in the learning ability, which is described as a newly added context under the strict definition in this paper, and it represents an innovation**. Due to the flexibility and indefinability of context in interactions with the world, AI may establish a non-human context, resulting in constraints failing and agency issues (essentially), because AI lacks the innate social nature of humans and constructs concepts differently from humans (Section 3).
>
> Therefore, the *title of this paper's position is: Position (AXMX)*: Rules Created by Symbolic Systems Cannot Constrain a Learning System (SB8V)
> 1.  There is a flaw in natural language, which we analyze in Section 2 (artificial symbols inherently have no meaning; Class-based Symbolic System; context is hard to define; for humans, context is not a choice but a new construction, and the interpretation of symbols belongs to society, thus allowing rules to function, etc.).
> 2. There is a misalignment in abilities (Sec 3;4;B;F;J;K)—the concept formation occurs in the perception of thinking symbols and thinking language. Natural language is the result of human-style perception and thinking, and fluent communication on the symbolic level is not equivalent to fluent communication on the imaginative level. This is also discussed in the Triangle Problem. The first Triangle Problem reflects the definition of concepts (symbolic stickiness), and the second Triangle problem reflects the use of concepts (generation and behavior under conceptual stickiness). ***A large number of examples are provided in lines 287, Appendix J,K and L.***

---

### Official Review · Reviewer_8rwE · 2025-03-14

**Significance:** 2
**Argument Clarity:** 2
**Rating:** 2
**Confidence:** 4

**Questions:**

I have a clarification question :
1. The author seems to be positioned against symbols. However, they do not differentiate between rule based constraints v/s more tacit - i.e. those being conveyed via data driven approaches. Do we care about this distinction in this position paper, or we stop the argument at "lost in translation". That is, the fact that symbols will never really capture the true meaning, it will impact both rule based constraints and data driven constraints - and therefore symbols are not useful? While I'm not sure of the author's take, I feel this distinction conveys the impact of using symbol for constraints - I would suppose the two strategies would present different kind of issues - which may lead us to realize whether using symbols in certain ways is better than certain other ways.

2. The paper advocates that symbols are lossy, however by the scope of what truly captures our "meaning" we want to convey, what media is lossless? I'd assume even if all the sensory information is captured throughout time (space time tubes in Kambhampati et al) - the mental model asymmetry still induces additional context which is arguably impossible to capture.

3. I agree with the sentiment of the position in that a human is involved in the communication process. What is the authors take on AI - AI interaction? Are symbols equally flawed when two AI systems are interacting and one tries to constrain the other? With humans, an irreducible component of why symbol will always be vague is the inability to perfectly mental model. But with AI agents we they can just share their entire code base / model weights to the other agent to share exactly what it "means".

**Discussion Potential:**

3

**Paper Summary:**

The paper advocates that the use of symbols to control / constraint AI is not enough primarily because of the inherent (possibly irreducible) ambiguity in the symbols (as tokens). While I was aware of several ways why this ambiguity is irreducible, the paper covers several additional reasons as to why symbols will always have grounding issue. This inability of symbolic structures is because symbols cannot capture the true "meaning" which would require space time tube + context + societal interactions of the human interacting + their cultural background and all possible context. Let's say it does (my argument may be absent in the paper, but agrees to the sentiment) there is intrinsic "intuitive" and tacit components of human mental model for which symbols are not enough. This grounding issue is used to take the position that AI systems - given that their principal mode is text / image etc. perception - cannot understand the true meaning just through symbols and therefore we can fail to constrain AI systems.

**Position:**

Yes

**Position In Title:**

Yes

**Related Work:**

3

**Strengths And Weaknesses:**

I agree with the position, simply put, that symbols are not enough to constraint AI systems.

But I have some issues :

1. There already exists arguments to show why symbols do not capture space time tube / entire context ever required to grasp the meaning. A trivial one being the lack of a perfect way of mental modeling the listener - which causes an irreducible error in grounding the meaning of a symbol. While the paper does a really good job of highlighting several other reasons (why symbols will always be vague), I wonder if my trivial argument is enough to suggest the position taken by the paper.

2.  While I admit that there is immense value in highlighting a problem without taking the onus of providing a solution, in this particular case, I feel that my trivial argument above is known to the community. Therefore, the main recommendation of the work "symbols don't guarantee constraints to AI systems" could already be derived. Can the author justify novelty - why this position was not known?

3. While symbols cannot guarantee lossless transfer of "meaning" because of the various factors inducing hidden, potentially tacit and infinitely large context - the paper doesn't touch on the topic of what alternatives do we have? Specifically, I would have liked to read - Is the community using symbols because that's our only choice, or because we were in misbelief on the shortcomings of using symbols.

4. Another offshoot of previous question is - we don't train LLMs with raw symbols. Each token / symbol is presented to the learning system in the context of sentences. While I agree that it's lossy, the paper doesn't highlight what areas of AI safety gets impacted the most.

Works like "Symbols as a Lingua Franca for Bridging Human-AI Chasm for Explainable and Advisable AI Systems" by Kambhampati et al already takes a position on that AI systems should make best effort on interacting with humans in the loop with symbols. While the paper maybe a good resource for the various shortcomings of symbols as constraints, I feel certain trivial ones provide good enough justification to the central argument of the position. Since the paper doesn't differentiate between the impact of various different shortcomings on AI safety or different ways of constraining AI systems, I'm unable to have a fine-grained understanding of the disadvantage of using symbols - beyond "symbols are lossy". I suggest the authors to consider leading the position paper from the perspective of impact caused by the various ways symbols are not a suitable choice and provide some feasible alternatives.

**Support:**

2

---

> ### Author Rebuttal · Authors · 2025-03-26
>
> I greatly appreciate your work, time, and the questions you've raised. I hope I can address your misunderstandings and concerns.
>
> 1. Issue-1: Is this trivial argument sufficient? The response is not adequate.
>
>      **The trivial argument has already been thoroughly discussed and explained in the main text (2.5 Path Media for Transmitting and Interpreting Imaginative Space, Natural language cannot fully reproduce the imaginative space).**
>
>     - Reason 1: **The aim of this paper is to provide a solid theoretical foundation and background for community discussions. We have considered nearly all possible aspects. Therefore, this is just one of the viewpoints presented in the paper.**
>     - Reason 2: More is needed to fully discuss the mechanisms of human language formation—specifically, how meaning is stabilized through the stickiness of symbols and concepts, which makes natural language consistent despite its flaws. AI does not possess human-like value knowledge.
> 2. I-2: Is this point already known to the community, or has previous work already been done on this? **No**
>    - **This paper is the first to propose this idea (Rules Created by Symbolic Systems Cannot Constrain a Learning System), providing a complete logical chain and rigorous definitions. This is precisely the investigation that led to the writing of this paper, and we emphasize in Sec4 that the community should not ignore this issue. We also discuss its essence in detail. This viewpoint has not been published by anyone else; we are the first to complete and finalize this work. Another major contribution is the definition of a new type of Principal-Agent Problem in economics, which we are also the first to complete, distinguish, and define.**
> 3. I-3, Q-2: Why no other solutions are provided, and what other solutions or lossless media exist?
>    1.  As discussed in GR, our goal is to show why it fails; the alternative views argue it works. Hence, no solution is required here.
>    2. **We hope to spark community discussion, which is why we submitted a position paper. In fact, we had already thought of and had solutions to your question**, namely, what medium is lossless—Split-Brain Syndrome. For normal people, the brain is a whole, but for split-brain patients, their brain is split into two distinct entities, which leads to different behaviors and views. (Gazzaniga, M. S., & Sperry, R. W., 1967). This is also the main reason we reject symbolic constraints, **implying**(line 308) the neural collective. **The subsequent community discussion and related research will be about how to convert symbolic rules into neural rules. (NG2z) The constraints we humans face are due to costs, from our natural sense of morality and self-respect, and societal punishment, so we create rules. However, how these rules are converted into neural language or structures (like the *prefrontal cortex*) or other methods is something we hope the community will discuss. (This paper is not a methods-position paper, *so we do not wish to reveal the answer prematurely to narrow the community's perspective*.)**
>         1. What kinds of constraints or conversion methods?
>         2. How much freedom or agency should we give AI?
>         3. Does AI have rights? Is AI conscious? Does this method violate ethics and morality?
>         4. If AI aligns with humans, does it lead to self-awareness (4.1)? If AI doesn't align with humans, can it replace humans(K)?
>
> 4. I-4: We explicitly state in the paper which areas of AI safety are most affected and provide a comprehensive and complete violation method and definition (Sec4.1, L). We do not go further to allow the community to work on this solid theoretical foundation and viewpoint.
>
>     1. For example, L3, L4 related to CBRN.
>     2. Verification Content 4 on LLM.
>     3. Sec3, 4.1, K, discussions on multimodality and superintelligence.
> 5. I-5: The work you recommend[will add to Alternative Views] is exactly the position we oppose—that consistency in symbolic behavior is insufficient. **This is precisely the issue we discuss in Sec3, where we explain why it doesn't work with the Triangle Problem (line244).**
> 6. Q-1: **In fact, we strongly emphasize data-driven methods, specifically updates to concepts that occur through interaction with the real world. As we mentioned in GR**, the paper discusses how AI forms concepts differently from humans because its innate knowledge differs, and how the interpretation of symbols is not controlled by society, thus modifying the human symbol system and developing its own (symbol) system and patterns(line 238,366).
>
> 7. Q-3: **This is formally discussed on the last page of the paper**, where we introduce the impossibility of this situation. It would lead to the absence of negotiation between humans and AI, and a rapid extension of a belief, leading to the swift formation of collective action. **This is also something the paper strongly calls for avoiding, as even a small error can result in huge consequences.**

---

### Official Review · Reviewer_NG2z · 2025-03-14

**Significance:** 3
**Argument Clarity:** 3
**Rating:** 2
**Confidence:** 2

**Questions:**

* A lot of the weaknesses contain parts of the question, and would help me understand what the authors mean, but concretely, take the motherland example from the paper: doesn'tthe confusion arise precisely because the context is different. Clearly the actors in the story would not have a completely fluent communication in the symbolic interaction space, because if they tried, they would identify a mismatch between their definitions. Wouldn't that be a good counterargument to using it as an example of the Triangle Problem?

**Discussion Potential:**

4

**Paper Summary:**

The paper argues that symbolic systems (e.g., laws, regulations, natural language) are fundamentally incapable of constraining autonomous learning systems like AI. It claims that symbols are not defined by their intrinsict properties, but extrinsic: they are defined by how they're used, the context they appear in, the the interpretation that society puts in them. They devise the Triangle Problem framework, and illustrate the disconnect between human and AI conceptual spaces, leading to risks such as symbolic jailbreaks and a novel principal-agent problem. The paper critiques existing approaches (e.g., Asimov’s laws, RLHF) and advocates for Symbolic Safety Science to address AI alignment challenges, emphasizing the need for frameworks that account for differences in concept formation and context interpretation.

**Position:**

Yes

**Position In Title:**

Yes

**Related Work:**

3

**Strengths And Weaknesses:**

### Strengths

* The paper Identifies a very useful signal that symbols themselves do not have meaning; but rather that  it is their use and the context they're a part of that assigns them meaning.
* The paper identifies the importance of understanding this statement in the context of modern day machine learning where we use symbols to constrain learning systems
* The paper suggests a concrete framework - the Triangle Problem - for understanding the nuance present here

### Weaknesses

* The paper asserts that symbolic systems cannot constrain learning systems due to their lack of intrinsic meaning. But as I see it, symbolic constraints are foundational to enabling learning in the first place. For example, with training data, symbols (e.g., labels, instructions) guide learning systems by providing extrinsic meaning through human-defined associations (e.g., "cat" as a label for images of cats). Without these symbols, supervised learning frameworks would collapse. Another example is reward functions: In reinforcement learning, symbolic rules (e.g., "maximize score") extrinsically define objectives. These constraints shape behavior even though the symbols themselves lack inherent meaning. From where I stand, it appears that the paper overlooks the duality of symbolis: they act as both enablers and constraints. Their extrinsic meaning, derived from context and training, is precisely what allows humans to steer learning systems toward desired outcomes. If the paper wishes to argue that symbols cannot constrain learning, then I see it necessary to argue why it is that they can enable it.

* The paper dismisses symbolic systems because they rely on extrinsic meaning, but to me this appears to be their strength, not a weakness. Take the example of social reinforcement: human language itself is a symbolic system that works because meanings are extrinsically stabilized by societal consensus. For instance, legal systems use symbols (laws) with no intrinsic meaning, yet they effectively constrain behavior through shared interpretation. Another example is AI alignment: techniques like RLHF extrinsically align AI with human values by encoding preferences into symbolic feedback. Symbols without context are indeed fluid, but we're only ever using them within existing contexts. Because of this, I'm not sure if I buy the argument that extrinsic meaning is inherently unstable

**Support:**

2

---

> ### Author Rebuttal · Authors · 2025-03-28
>
> We sincerely appreciate your time and the comments you’ve provided. We will address your concerns one by one, **clarifying how they stem from misunderstandings of the main arguments and also highlighting the unique contributions of this paper**.
>
> # 1. Training or Updating? The Issue Lies Not in Training, But in Updating. (Weak1，Weak2)
>
> Before diving into the specifics, we kindly ask you to first review the GR section to better understand the intent and framework of our paper.
>
> First, we would like to clarify that what **we are discussing concerns the conceptual updating and evolution of AI in real-world activities as the autonomy of AI agents increases**. This leads to a failure of pre-established rules (The rationale for establishing such rules is marked in [8rwE, 3-2]). This is also emphasized in our first introduction of the concept of the New Principal-Agent Problems within the field of economics (see line 419). **Although we only explicitly state in the article that we are the first to propose this position and the Triangle Problem framework, the paper also presents several other original contributions, as noted in responses [SB8V, MP3].**
>
> ## **Therefore, this is a highly interdisciplinary work that integrates cognitive science, linguistics, artificial intelligence, and economics—rather than a paper confined to a single technical domain—because the issues raised by this position cannot be resolved within any single field alone.**
>
> First, we fully agree with your point in Weak1 regarding training. ***However, the focus of this paper is on the problem of concept evolution and updating that occurs after training, when the AI interacts with the real world. It is this updating (learning ability) that causes the symbolic system to fail in constraining the learning system.***
>
> **The key issue here is Symbolic Stickiness and Conceptual Stickiness, i.e., the meanings of symbols evolve as contexts change. Humans have an innate baseline to determine what kinds of learning and concepts are valid and acceptable. We describe this as innate knowledge (innate organs, value knowledge—including innate evaluation and intuition, see Sec. 3). Furthermore, the right to interpret symbols does not belong to individuals but to society. This is something AI fundamentally lacks.**
>
> ***Next, I will address another point of confusion: Why doesn’t social reinforcement work? This is precisely the issue described throughout the paper. Social constraints are grounded in human nature and the survival-driven sense of reasonableness (Sec. 4). The essence of social constraint is cost, as discussed in [8rwE, 3-2].***
>
> In Sec. 2, we introduce the characteristics of human language and emphasize that the function of context is not determined by individuals, but by society (line 137, R). It is also emphasized in line 1048 and line 1166.
>
> In Sec. 3, we introduce the process by which the world and Innate Knowledge contribute to the formation of concepts and language—with concept formation referred to as Thinking Symbol and thought formation as Thinking Language. These are the results of humans perceiving the world through their cognitive abilities, grounded in inherent human capacities (E，F，J，K). Through highly similar genetic inheritance and shared environmental conditions, a common set of perceptual selections emerges, giving rise to their external expressions: Symbol and Language.
>
> Natural language is the outer shell of human thought (how we understand and view the world). It arises from the raw materials of concept generation provided by the world:$\begin{bmatrix}Encounterable  \\\\  Observable  \\\\  Describable  \\\\  Definable  \\\\  Distinguishable  \\\\  Operable\end{bmatrix}$ (line 1032), resulting in  $Concepts=\begin{cases}Objects  \\\\  Relations  \\\\  Actions  \\\\  Systems  \\\\  Environments  \\\\  Scopes  \\\\  Dimensions  \\\\  Dimension-Values  \\\\  Capabilities  \\\\  Correlations\end{cases}$ (line 1013). This is also the first time these have been clearly defined and explained, which significantly enriches and complements PSS theory.
>
> ## 2. Question 1:
>
> The "Motherland problem" emphasizes that even human agents—who are aligned in capabilities—can form erroneous concepts due to differences in their worlds, and such misconceptions often remain uncorrected in most scenarios. AI and humans differ drastically in both their worlds (including social roles and material interactions) and capabilities.
>
> As emphasized in our response to (8rwE). alignment at the symbolic level alone is far from sufficient. This leads directly to the Triangle Problem 2, resulting in major behavioral divergence in XY (i.e., behavior patterns). Such divergences are essentially rooted in human social concepts and empathic perception—outcomes of humans being deeply embedded in the world. A more detailed explanation can be found in the (line 309), particularly under the four Verification Contents, with Verification Content 4 offering a focused discussion.

---

### Decision · Program_Chairs · 2025-04-27

**Decision:**

Reject

**Comment:**

This paper takes the position that symbolic systems cannot effectively constrain learning-based systems because symbols lack intrinsic meaning and drift semantically over time. It introduces a new framework, the “Triangle Problem,” to explore this problem and proposes a new subfield of “Symbolic Safety Science.”

Whiel the reviewers did appreciate the premise, there were concerns about the paper's claims and how it tried to support them. For example, Reviewer NG2z appreciated the core idea (“symbols do not have meaning; it is their use and context that assign them meaning”) but pointed out a key issue that “If symbols can’t constrain learning systems, why do they so reliably enable them during training?”

Reviewer AXMX said, “I just fail to understand,” which was a sentiment shared by another reviewer. The paper was made more difficult to understand because of undefined jargon. For example, a reviewer noted that "The paper is in general hard to follow, there is no clear definition of many terms, also not of those introduced in this paper (e.g., the triangle problem)." SB8V raised concerns about novelty, describing the paper as taking "a view that does not strike [the reviewer] as novel.” Others, like 8rwE, asked why the proposed critique of symbolic alignment was not already obvious from earlier grounding problem literature, saying " I feel that my trivial argument above is known to the community."

The rebuttal offered detailed responses, but many reviewers felt it reiterated rather than clarified. The paper may be raising important points, but it doesn’t engage cleanly with existing literature or offer the kind of clear position and actionable implications we expect from an ICML position paper. As one reviewer put it: “It is a very philosophical AI paper, following up on the symbol grounding discussion, which is going on since Harnad's seminal paper of 1990. There is no discussion on any practical or theoretical ramifications for machine learning. I believe that this would already be sufficient reason for rejecting this paper."